# Dissolved organic carbon, major and trace elements in peat pore water of sporadic, discontinuous and continuous permafrost zone of Western Siberia

Tatiana V. Raudina[1], Sergey V. Loiko[1], Artyom G. Lim[1], Ivan V. Krickov[1], Liudmila S. Shirokova[2,3], Georgy I. Istigechev[1], Daria M. Kuzmina[1], Sergey P. Kulizhsky[1], Sergey N. Vorobyev[1], Oleg S. Pokrovsky[2]*

[1] BIO-GEO-CLIM Laboratory, Tomsk State University, Lenina av., 36, Tomsk, Russia
[2] Geoscience and Environment Toulouse, UMR 5563 CNRS University of Toulouse (France), 14 Avenue Edouard Belin, 31400 Toulouse, France
[3] N. Laverov Federal Center for Integrated Arctic Research, Russian Academy of Science, Arkhangelsk, Russia

*Correspondence to*: Oleg S. Pokrovsky (oleg.pokrovsky@get.omp.eu)

**Abstract.** Mobilization of dissolved organic carbon (DOC) and related trace elements (TE) from the frozen peat to surface waters in the permafrost zone is expected to enhance under on-going permafrost thaw and active layer thickness (ALT) deepening in high latitude regions. The interstitial soil solutions are efficient tracers of on-going bio-geochemical processes in the critical zone and can help to decipher the intensity of carbon and metals migration from the soil to the rivers and further to the ocean. To this end, we collected, across a 640-km latitudinal transect of sporadic to continuous permafrost zone of western Siberia peatlands, soil porewaters from 30-cm depth using suction cups and we analyzed DOC, DIC and 40 major and TE in 0.45-µm filtered fraction of 80 soil porewaters.

Despite an expected decrease of the intensity of DOC and TE mobilization from the soil and vegetation litter to the interstitial fluids with the increase of the permafrost coverage, decrease in the annual temperature and ALT, the DOC and many major and trace element did not exhibit any distinct decrease in concentration along the latitudinal transect from 62.2°N to 67.4°N. The DOC demonstrated a maximum of concentration at 66°N, on the border of discontinuous/continuous permafrost zone, whereas the DOC concentration in peat soil solutions from continuous permafrost zone was equal or higher than that in sporadic/discontinuous permafrost zone. Moreover, a number of major (Ca, Mg) and trace (Al, Ti, Sr, Ga, rare earth elements (REEs), Zr, Hf, Th) elements exhibited an increasing, not decreasing northward concentration trend. We hypothesize that the effect of temperature and thickness of the ALT are of secondary importance relative to the leaching capacity of peat which is in turn controlled by the water saturation of the peat core. The water residence time in peat pores also plays a role in enriching the fluids in some elements: the DOC, V, Cu, Pb, REE, Th were a factor of 1.5 to 2.0 higher in mounds relative to hollows. As such, it is possible that the time of reaction between the peat and downward infiltrating waters essentially controls the degree of peat pore-water enrichments in DOC and other solutes. A two-degree northward shift in the position of the permafrost boundaries may bring about a factor of 1.3 ± 0.2 decrease in Ca, Mg, Sr, Al, Fe, Ti, Mn, Ni, Co, V, Zr, Hf, Th and REE porewater concentration in continuous and discontinuous permafrost zones, and a possible decrease in DOC, SUVA, Ca, Mg, Fe and Sr will not exceed 20% of their current values. The projected increase of ALT and vegetation density, northward migration of the permafrost boundary, or the change of hydrological regime are unlikely to modify chemical composition of peat pore water fluids larger than their natural variations within different micro-landscapes, i.e., within a factor of 2. The decrease of DOC and metal delivery to small rivers and lakes by peat soil leachate may also decrease the overall

export of dissolved components from continuous permafrost zone to the Arctic Ocean. This challenges the current
paradigm on the increase of DOC export from the land to the ocean under climate warming in high latitudes.
**1 Introduction**
Boreal and subarctic regions of the Northern Hemisphere are among the most vulnerable areas to on-going
climate warming (Natali et al., 2011, 2015; Schuur et al., 2015; Vonk et al., 2015b; Pries et al., 2016). Because of
sizeable carbon storage in frozen soils of Siberia (Botch et al., 1995; Krementski et al., 2003; Frey and Smith, 2007;
Beilman et al., 2009; Tarnocai et al., 2009; Gentsch et al., 2015), the warming in this region is especially important for
global projections of the carbon balance on the planet (Smith et al., 2004; Frey and Smith, 2005; Feng et al., 2013). In
this regard, permafrost-bearing part of Western Siberia Lowland (WSL) is highly sensitive to soil warming, due to (*i*) the
dominance of discontinuous, sporadic and intermittent permafrost coverage compared to continuous and discontinuous
permafrost of central and eastern Siberia and Canada High Arctic; (*ii*) the surface layer temperature of the WSL
permafrost is often between 0 and -2°C, which is warmer than in other regions of the world (Romanovsky et al., 2010);
(*iii*) essentially flat area of the WSL and high impact of flooding and thermokarst development, and most importantly (*iv*)
high stock of ancient and recent organic carbon in the form of partially frozen peat deposits of 1 to 4 m thickness.
Mobilization of dissolved organic and inorganic carbon (DOC and DIC, respectively) and related trace elements
(TE) including metal contaminants and micronutrients from the frozen peat to surface waters and further to the Arctic
Ocean is one the major consequences of on-going permafrost thaw (Tank et al., 2012a, b, 2016; Striegl et al., 2005;
Rember and Trefry, 2004; Prokushkin et al., 2011; Mann et al., 2012; Grosse et al., 2016; Holmes et al., 2013). The
impact of warming on arctic and subarctic soil is primarily through the active layer thickness (ALT) rise (Zhang et al.,
2005; Akerman and Johannson, 2008) although a number of other phenomena (plant productivity, drainage and
hydrological regime change, ground fires etc) may be even more important in changing the biogeochemical cycle of
carbon and metals in permafrost-affected soils (Jorgenson et al., 2013). For these reasons, the peat land zones have
received significant attention (Haapalehto et al., 2011; Olefeldt and Roulet, 2012; Charman et al., 2013; Quinton and
Baltzer, 2013; Muller et al., 2015; Morison et al., 2017), notably via natural manipulation experiments in order to assess
the responses of peat carbon to simulated warming and oxidizing (Dielman et al., 2016; Liu et al., 2016), water table
manipulation (Blodau and Moore, 2003; Strack et al., 2008; Goldberg et al., 2010) and drought (Clark et al., 2012).
The majority of available studies adressed the carbon and element transformation in the permafrost regions via
analysis of rivers (Lobbes et al., 2000; Striegl et al., 2005; Spencer et al., 2008, 2015; Holmes et al., 2012; Wickland et
al., 2012; Giesler et al., 2014; Mann et al., 2015), lakes (Kokelj et al., 2005, 2009; Guo et al., 2007; Laurion et al., 2010;
Tank et al., 2009), mires (Olefeldt and Roulet, 2012; Olefeldt et al., 2013, 2014) or soil organic matter (SOM) from
various depth and soil aqueous leachate (Swindles et al., 2015; Hodgkins et al., 2014, 2016; Drake et al., 2015; Vonk et
al., 2015a; Yang et al., 2016) and largely ignored soil porewater chemistry. At the same time, interstitial soil solutions are
known to be efficient tracers of on-going bio-geochemical processes in the critical zone (Hendershot et al., 1992; Stutter
and Billett, 2003; Quinton and Pomeroy, 2006; Karavanova and Malinina, 2007; Gangloff et al., 2016) and can help to
decipher the intensity of carbon and metals migration from the soil to the rivers and further to the ocean. However, in
contrast to significant number of in-situ measurements of DOC and metals in the interstitial soil solutions of the boreal
zone (Van Hees et al., 2000a, b; Reynolds et al., 2004; Starr and Ukonmaanaho, 2004; Michalzik et al., 2001; Giesler et
al., 2006; Ilina et al., 2014; Griffiths and Sebestyen, 2016; Shotyk et al., 2016) there are relatively few studies of soil
porewaters from the permafrost regions (e.g., Marlin et al., 1993; Prokushkin et al., 2005; Pokrovsky et al., 2006, 2013;

Koch et al., 2013; Jessen et al., 2014; Fouche et al., 2014; Fouché et al., 2014; Mavromatis et al., 2014; Herndon et al., 2015), none of them dealing with organic-rich peatland soils. Only recently, Frey et al. (2016) reported results soil pore waters from the yedoma wetland soil within the flow-path continuum from the soil to the Kolyma River mainstream.

In this work we sampled, across a 640-km latitudinal transect of sporadic to continuous permafrost, the interstitial soil solutions of the largest peatland of the world. Our main goal was to quantify the distribution of DOC, major and TE in pore waters along a permafrost gradient of similar micro-landscapes. Within the upper unfrozen peat horizon, we hypothesize a trend of diminishing DOC and metal concentration northward, due to the decrease of mean annual temperature, vegetation density and active layer thickness. We aimed at quantifying the latitudinal trend of peat pore water concentration of DOC, major and TE and testing the difference in solute concentration sampled from various micro-landscape such as mound, hollow, depression, and polygon. Implying a substituting-space-for-time approach, developed for surface waters of western Siberia, (i.e., Frey et al., 2007a, b; Frey and Smith, 2005), the obtained results should allow a straightforward empirical provisions of soil water chemistry change during northward migration of the permafrost boundary. Because the main source to inland waters in this vast territory (over 1 million km²) occurs as supra-permafrost flow over the impermeable frozen peat horizon (Novikov et al., 2009), and due to the fact that the West Siberian peatlands contain the largest soil water and ice resources in the northern hemisphere (Smith et al., 2012), the assessment of soil peat water chemical composition should help predicting the possible change of DOC and metal transport of permafrost-bearing Siberian rivers and lakes under climate warming scenarios.

## 2. Materials and Methods

### 2.1. Geographical setting and local micro-landcapes

Western Siberia Lowland (WSL) includes the watershed of the Ob, Pur, Nadym, Poluy and Taz rivers that drain Pleistocene sands and clays, covered by thick (1 to 3 m) peat. All three major zones of the boreal biome, taiga, forest-tundra and tundra, can be found in this region. The territory investigated in this work includes 3 main permafrost zones: sporadic, discontinuous and continuous (Fig. 1). Quaternary clays, sands, and alevrolites underlying the surface peat deposits range in thickness from several meters to 200-250 m and have fluvio-glacial and lake-glacial origin in the north of 60°N. The climate is humid semi-continental with mean annual temperature (MAT) ranging from - 2.8°C in the south of the cryolithozone (Syrgut region) to -9.1°C in the north (Tazovsky). The annual precipitation ranges from 600 mm in Kogalym to 360 mm in Tazovsky. Along the gradient of discontinuous to sporadic to continuous permafrost zone, we selected 5 main test sites whose physico-geographical characteristics are given in **Table 1.**

A typical feature of the WSL is the presence of positive and negative forms of relief – microlandscapes. The initial bog with weakly pronounced micro-relief was subjected to freezing during Subboreal period (~ 4500 y.a.). During Subatlantic period (2500 y.a.) and the increase of temperature and precipitation, the thermokarst started. The hollows received sufficient water and they started to thaw, whereas the mounds were rising due to ice wedges underneath (Panova et al., 2010; Ponomareva et al., 2012; Pastukhov et al., 2016). The positive forms include ridges in permafrost-free and sporadic permafrost zone, mounds in discontinuous permafrost zones, and polygons in the subarctic tundra of continuous permafrost. The negative forms comprise hollows (abundant across all zones), permafrost subsidences in discontinuous and continuous permafrost zones, and frost cracks of the polygonal tundra biome. In each of five major sites, several micro-landscapes corresponding to one positive and two negative form of relief were selected as specified in **Table 1** and shown as aerial views in **Fig. 1.** The cross sections of dominant micro-landscapes with corresponding soil specifications

are represented in **Fig. 2** and include: (*i*) peat mounds in the 4 southern sites of flat mound peat bog, and corresponding
polygon in the most northern, Tazovsky site of polygonal tundra; (*ii*) hollows in all 5 sites, and (*iii*) permafrost
subsidences in 4 southern sites and corresponding frost crack in Tazovsky. Typical soil profiles of studied sites are
illustrated in **Fig. S1** of Supplement.

**2.2. Soil porewater sampling**
Altogether, 80 soil porous waters in 5 main sampling sites were collected in the end of July-beginning of August 2015. In
this study, suction cup lysimeters were used. The chemical composition of interstitial soil solution is known to depend on
the extraction method (e.g., Geibe et al., 2006; Schlotter et al., 2012). Detailed comparison between suction cup and press
technique is described in methodological work of our group (Raudina et al., 2016). In the peat profile of each
microlandscape, the PTFE suction cup lysimeters (95 mm long and 21 mm diameter, 2 μm pore size) of SDEC (France)
were installed at the depth of 30±15 cm below the moss layer (**Fig. S2** of Supplement). The choice of the sampling depth
was determined by the position of the permafrost table: typically, the cup was installed at 10 cm from the peat outcrop
vertical surface, 5-10 cm above the bottom of the active layer, but not deeper than 40-50 cm from the moss layer. In all
sites, the cups were installed exclusively in soils that belonged to group Histosols (according to WRB 2014, i.e., having a
thickness of peat > 60 cm). The cups were connected via PTFE tubing to polypropylene 1-L container maintained at 75
to 50 kPa via a Mityvac MV8255 PVC-made hand pump or a portable electric vacuum pump (KNF Neuberger W/VAC.
5.5 L). Before each installation, the suction cups were cleaned by flushing with Milli-Q water (~ 250 mL), followed by
3% ultrapure HNO3 (~ 250 mL) and finally Milli-Q water (~ 750 mL). Each cup was soaked in Milli-Q water for at least
1 day before the experiment and was used only once. The porewater was collected in two steps. The first portion (100-
200 mL) was collected during 24 h and the fluid was discarded, allowing for the saturation of the tubing and the recipient
bottle surface. The $2^{nd}$ portion (100-300 mL) was collected during the next 24 h of deployment or, in case of dryer
conditions, over 48 h and used for analyses. The vacuum in the recipient bottle decreased from 75 kPa to atmospheric
pressure over 24 h, and the first portion of the fluid appeared at 45 to 50 kPa.

**2.3. Analyses**

Collected waters were immediately filtered in pre-washed 30-mL PP Nalgene® flacons through single-use

Minisart filter units (Sartorius, acetate cellulose filter) having a diameter of 25 mm and a pore size of 0.45 μm. The first
20 mL of filtrate were discarded. Filtered solutions for cation analyses were acidified (pH ~ 2) with ultrapure double-
distilled $HNO_3$ and stored in pre-washed HDPE bottles. The preparation of bottles for sample storage was performed in a
clean bench room (ISO A 10,000). Blanks were performed to control the level of pollution induced by sampling and
filtration. The DOC blanks of MilliQ filtrate never exceeded 0.1 mg/L which is quite low for the organic-rich pore waters
sampled in this study (i.e., 10–100 mg/L DOC). pH was measured in the field using a combined electrode with un
uncertainty of ±0.02 pH units. DOC and DIC were analyzed using a Carbon Total Analyzer (Shimadzu TOC VSCN)
with an uncertainty better than 3%. The instrument was calibrated for analysis of both form of dissolved carbon in
organic-rich, DIC-poor waters (e.g., Prokushkin et al., 2011). The UV absorbance of the filtered samples was measured
at 280 nm using quartz 10-mm cuvette on Cary-50 spectrophotometer. The specific UV-absorbency ($SUVA_{280}$, L $mg^{-1}$ $m^{-1}$
$^{1}$) is used as a proxy for aromatic C, molecular weight and source of DOM (Uyguner and Bekbolet, 2005; Weishaar et al.,
2003; Ilina et al., 2014 and references therein). The $SUVA_{280}$ in the present study was used for consistency with previous
measurements of lakes and rivers in western Siberia (Shirokova et al., 2013; Manasypov et al., 2015, 2017; Pokrovsky et
al., 2015) and permafrost-draining rivers in Central Siberia (Prokushkin et al., 2011).
Major anions ($Cl^-$, $SO_4^{2-}$) concentrations were measured by ion chromatography (HPLC, Dionex ICS 2000) with
an uncertainty of 2%. Major cations (Ca, Mg, Na, K), Si and ~40 TE were determined with an ICP-MS Agilent ce 7500
with In and Re as internal standards and 3 various external standards, placed each 10 samples in a series of river water.
Details of TE analyses in DOC-rich waters of western Siberia are given elsewhere (Pokrovsky et al., 2016a, b). The
SLRS-5 (Riverine Water Reference Material for Trace Metals certified by the National Research Council of Canada) was
used to check the accuracy and reproducibility of each analysis (Yeghicheyan et al., 2013). Only the elements that
exhibited good agreement between replicated measurements of SLRS-5 and the certified values (relative difference <
15%) are reported in this study.

**2.4. Statistical treatment**
The concentrations of carbon and major elements in soil porewaters were treated using the least squares method and
Pearson correlation (SigmaPlot version 11.0/Systat Software, Inc). Regressions and power functions were used to
examine the relationships between the elemental concentrations and the latitude of sampling. The normality of data
distribution was checked using the criterion of Kolmogorov-Smirnov, separately for each site and for the full set of the
data. The significance value was < 0.01 and thus non-parametric criteria for data comparison were used. First, major and
TE concentrations in soil porewaters of (1) five main sampling sites and (2) four main micro-relief landscapes (polygon,
permafrost/subsidence, frost crack and hollow) were processed using non-parametric H-criterion Kruskal–Wallis test.
This test is suitable for evaluation of difference of each component among several samplings simultaneously. It is
considered statistically significant at $p < 0.05$. In case of significant differences, a comparison of DOC, major and TE
concentration between soil porewaters sampled in 3 main pair micro-landscapes (mound-hollow, mound-subsidence, and
hollow-subsidence) of each 5 major sampling site was conducted using non-parametric pair Wilcoxon-Mann Whitney
test. All graphics were performed using MS Excel 2010 and GS Grapher 11 package. Principal component analysis
(PCA) was used for the full set of sampled soil porewaters across the micro-landscapes and permafrost zones. In this
treatment, the main numerical variables were the geographic latitude of the sampling site, the depth of peat horizon,
ALT, specific conductivity, pH, DOC, DIC, Cl, $SO_4$, Si, all major cations and 43 TE concentration.
The PCA analysis allowed to test the influence of various parameters, notably the latitude and the ALT on the soil
porewater DOC and element variability. All the variables were normalized as necessary in standard package of
STATISTICA-7 (http://www.statsoft.com) given that the units of measurements of various components are different.
The identification of factors was performed using the method of Raw Data and the extraction method was principal
component. The scree test involved plotting the eigenvalues in descending order of their magnitude against their factor
numbers and determining where they level off. The PCA values demonstrated significant decrease of the value between
F2 and F3 suggesting therefore that at least two factors are interpretable.


**3. Results**
**3.1. PCA analysis and correlations between elements**
The PCA analysis of all micro-landscapes and geographical zones yielded 2 possible factors contributing to
observed variations in element concentration (i.e., 20 and 9%, **Fig S3 (A, B)** of Supplement). Such relatively low

proportion of the variance explained by PCA is consistent with previous treatments of the WSL river water, conducted on a much larger dataset (Pokrovsky et al., 2016a). Because the standard STATISTICA-7 package used in this work does not allow realization of Kaiser-Meyer-Olkin (KMO) criterion, we computed this criterion using Excel®. The KMO value was equal to 0.533 which suggests rather low adequacy: the analysis does not make sense at KMO < 0.5. Note that the removal a part of the data series and conducting separate PCA for major elements, TE, various forms of micro-relief and various geographical sites did not yield any better description of the variance mainly because of insufficient size of the dataset.

The first factor explains a greater variance in heavy element hydrolysates such as REEs, Cr, Nb, Zr, Hf, Th and U whereas the second factor was pronounced for soluble and biogenic elements (Mn, Co, Ni, V, Si, Ca, Mg, Sr), pH and latitude but also included Al and Fe, presumably due to organic complexation (see section 4.2 below). The correlation matrix (**Table S1** of Supplement) and respective dendrogram of a hierarchical cluster for scaled pore water score variation (**Fig. S3 C**) demonstrated pronounced link of Si with REEs, Zr, Nb, Fe, Cr, V and Li, probably corresponding to the source of these elements from silicate matrix of the peat profile. There was positive correlation between Mn and Ca and Sr and Ca, reflecting the biological impact or soluble carbonate minerals as it is established for riverwater of the region (Pokrovsky et al., 2016a). Note that the correlations of latitude, specific conductivity, pH and DOC with all major and TE were poorly pronounced (R < 0.5), whereas Fe and Al correlated with Si, Ti, V, Cr, Co, Ni, As, Zr, heavy REE, Hf.

**3.2. Effect of micro-landscape**

The mean values with S.D. of all major and TE in soil porewaters of main microlandscapes in each site are listed in **Table 2**. The mean values for the whole WSL territory for two dominant micro-landscapes, mound and hollow, are given in the last two columns of this table. Results of the application of Wilcoxon-Mann Whitney test for assessing the differences of DOC and several major and TE mean values between the dominant micro-landscapes in each site are listed in **Table S2** of **Supplement**. According to the chosen statistical criteria, only a few elements (DOC, Al, Fe, Si, Mn, Cu, Cd, Pb, Hf, U) depicted significant differences in their concentration between different micro-landscapes. The DOC was approximately twice higher (p = 0.023 to 0.043) in mounds (or polygons) compared to hollows in all 4 sites except Pangody, where the difference was only a factor of 1.1 which is not significant (p = 0.082). In Khanymey, Urengoy and Tazovsky, the order of DOC concentration in various micro-landscapes was (mound or polygon) ≥ (permafrost subsidence or frost crack) > hollow. Cu and, sometimes, Zn, followed this order. Concentrations of Al, Si, Fe, Sr did not demonstrate any systematic difference between positive and negative forms of relief for each site, without distinct preferential enrichment of one microlandscape versus another in the north or in the south. The minimal contrast in DOC and element concentration between micro-landscapes was observed in Pangody and the maximal variability was in Khanymey.

Within the standard deviation of the mean values, there was no difference in DIC, Si, Ca and Mg concentration between different micro-landscapes in all studied sites. The exception was Khanymey where the hollows demonstrated a factor of 1.5-2.8 higher Mg, Si and Ca concentration compared to mounds and Urengoy where the mounds contained less Mg and Si than the hollows. However, in the latter case, at p = 0.041 to 0.048, this difference was within the variation of the average (**Table S 2**). The mean concentrations of DIC, Cl, K, Si, Ca, Mg, Al, Fe, Ti, Sr, Ba, Zn, Mn, Ni and TE over the full WSL territory are quite similar (±20%) between positive and negative forms of relief (compare the last two

columns of Table 2). The DOC, B, Na, V, Ga, Cu, Cs, Pb, REE and Th exhibited a factor of 1.5±0.2 (significant at p <
0.05) higher WSL-mean concentrations in mounds/polygons compared to hollows.

**3.3. Effect of latitude and permafrost zone on peat porewater concentrations of DOC and metals**
In order to examine the latitudinal trend of element concentration in the porewater, first we run the Kruskal-
Wallis and then the Wilcoxon-Mann Whitney pair test for overall differences. After that we assessed, which micro-
landscape exhibited the largest difference between sites. Results include the p-value of the difference between one given
site and other sites located northward (**Table S3** of the Supplement). The difference between sites was tested for
mounds/polygons and hollows for all 5 sites and for permafrost subsidence/frost crack for 3 most northern sites
(Khanymey, Urengoy and Tazovskiy). The DOC and major elements (Ca, K, Al, Si, Fe) exhibited clear difference (p <
0.05) between different geographic zones. The most pronounced difference between pair sites was observed for hollows.
Thus, the porewaters from hollows in most southern site (Kogalym, of the sporadic permafrost) demonstrated statistically
significant differences in DOC, Ca, K, Al, Si, Ni, Cu, Sr, Rb concentrations from hollows of Khanymey, Pangody,
Urengoy, and Tazovskiy. Among the elements listed in Table 2, DOC, Ca, Fe and Sr were found to be most sensitive to
the latitude of the sampling site regardless of the type of micro-landscape.
The general latitudinal trend in element concentration together with mean values in each micro-landscape as a
function of latitude was examined for all major and TE. The latitudinal trend was approximated by a linear regression
using all micro-landscapes and individually for hollows and mound/polygons:

$[\text{Element}] = A + B \times \text{Latitude (°N)}$                                    (1)

where $A$ and $B$ are the element-specific empirical coefficients. Parameters of equation for each element are listed in
**Table 3**. For most major components including DOC there was no systematic trend of increasing or decreasing of
average concentration across the 640 km latitudinal profile. There was a local maximum of DOC concentrations in
porewaters of peat mounds sampled at the Khanymey-Urengoy sites. Overall, 3 patterns of concentration – latitude
dependence could be distinguished shown in **Figs. 3-5** and **S4-S5**:
(1) Specific Conductivity, pH, DIC, DOC, K, Na, SO4, Si, B, Li, Fe, Ti, Cr, Ba, Mo, As, light REEs (La, Ce), W, and U
did not exhibit any statistically significant trend ($R^2 < 0.5$) or this trend was within the uncertainties as illustrated in **Fig.**
**3 A-H** and **Fig. S4 E-K**;
(2) A clear trend of steady increasing concentration northward was observed for $\text{SUVA}_{280}$, Mg, Ca, Al, Cu, V, Mn, Ni,
Sr, heavy REEs, Zr, Hf, Th ($0.45 < R^2 < 0.62$, $p < 0.05$). The overall increase from sporadic to continuous permafrost
zone ranged from a factor of 2 to a factor of 5, illustrated in **Fig. 4 A-H** and **Fig. S5 A-F**.
(3) Cl, Sb, Pb, Cd, Zn, Rb, and Cs exhibited a decreasing trend northward shown in **Fig. 5 A-E** ($0.48 < R^2 < 0.84$).
For some elements, there was a lack of any trend between 62°N and 66.5°N, followed by an increase (significant at p <
0.05) between 66 and 67.5°N: Ca (**Fig. 4 C**), Mn (**Fig. S5 A**), Co (**Fig. S5 B**), V (**Fig. 4 F**) and As (**Fig. S4 H**). The most
pronounced trend of element concentration increase northward was observed in mounds/polygons for Al ($R^2 = 0.91$), Sr
($R^2 = 0.69$), Zr ($R^2 = 0.57$), Ce ($R^2 = 0.76$), Hf ($R^2 = 0.68$) and Th ($R^2 = 0.92$). For these elements, the trend in
hollows/cracks was much less pronounced or even absent, with $R^2 < 0.5$ (**Table 3**). A decreasing trend of element
concentration northward was also better pronounced in mounds/polygons for Na, Cl, Rb, Cs and Pb.

**4. Discussion**

**4.1. Dissolved organic carbon transport in peat soils**

The first unexpected result of this study was the lack of significant decrease of DOC concentration in peat porewaters
northward, from sporadic to discontinuous and continuous permafrost zone (**Fig. 3 C**). The character of the DOM also
remained highly constant across the latitudinal / permafrost gradient as the $SUVA_{280}$ ranged from 2.4 to 3.5 L mg$^{-1}$ m$^{-1}$ in
all sites regardless of the microlandscape, with weak increase northward (**Fig. 4 A**). These values of $SUVA_{280}$ are
consistent with those of the lakes (2 to 4 L mg$^{-1}$ m$^{-1}$, Manasypov et al., 2015) and rivers (2 to 3.5 L mg$^{-1}$ m$^{-1}$, Pokrovsky
et al., 2015) of the region during summer period. The previously published values of $SUVA_{280}$ in WSL surface waters
were similar across a large scale of lake size (from 50 to 500,000 m²) and latitudinal position of the river watershed (from
57°N to 66°N). This strongly suggests highly uniform feeding of Siberian inland waters by allochthonous DOM
originated from peat leaching within the soil profile. The DOC transport to the river and lake presumably occurs via
suprapermafrost flow over the frozen peat layers at the depth ranging between 20 and 80 cm depending on the season, the
latitude and the micro-landscape context (see Fig. 2).  Given the similarity of $SUVA_{280}$ values across significant
geographical transect on positive forms of micro-relief (**Fig. 4 A, Table 3**), we hypothesize the similarity of the nature of
water-soluble OM that constitutes the peat on mounds. At the same time, sizeable increase in the $SUVA_{280\ nm}$ northward
may indicate a higher aromaticity of soil porewater DOM in the continuous permafrost zone relative to discontinuous and
sporadic permafrost zone (Fig. 4 A). The change of SUVA from 2.4 to 3.4 in hollows demonstrates a significant shift in
the composition of the DOM and may have a pronounced effect upon the biogeochemical processing of DOM upon
export as it has been recently shown in Eastern Siberia (Frey et al., 2016). This contradicts the conclusion reached in
recent studies of surface waters and soil leachates that the DOM leached from the permafrost soil layer has a consistently
lower concentration of aromatic carbon (i.e. lower $SUVA_{254}$ values, Mann et al., 2012; Cory et al., 2013, 2014; Abbott et
al., 2014; Ward and Cory, 2015), compared to DOM draining from the active, organic surface layer. However, the
majority of previous studies dealt with non-peat permafrost environment. In the case of the WSL peatland, the
contribution of UV-transparent microbial exometabolites and plant exudates including low molecular weight organic
acids  (i.e., Giesler et al., 2006) is certainly much higher in the southern forest-tundra and taiga zone compared to
northern sites of the polygonal tundra. In the present study, statistically significant increase of $SUVA_{280}$ northward in
hollows ($R^2$ = 0.599, see Table 3) may also indicate the lower rates of DOM processing in soils in the north, linked to
either shorter residence time of soil fluids or weaker processes of photo- and bio-degradation in continuous permafrost
zone compared to sporadic and discontinuous zone.
Generally higher DOC concentration in porewaters of mounds compared to that of hollows (Table 2) has two
possible explanations. The soluble DOC retainment by clay horizon  that underlays the peat in the WSL was
hypothesized as the main regulator of the DOC level in rivers of large latitudinal transect of WSL, from permafrost-free
to continuous permafrost zone (Pokrovsky et al., 2015). The gradient consisted in increasing the DOC concentration
northward of 64°N (Pokrovsky et al., 2015) because the DOC-adsorbing clay horizon that underlays the peat may be
frozen in the north (Kawahigashi et al., 2004). The latter authors suggested that the DOC in northern, permafrost-affected
tributaries of the Yenisey River was less biodegradable (and thus better preserved during its transport from the soil to the
river) than that in southern tributaries. If true, the lower DOC concentrations in hollows and subsidence relative to the
mounds observed in the present study is due to DOC adsorption on unfrozen mineral layers (silt, clays) located below the
peat horizon in depressions and hollows, which have much deeper position of the ALT than the mounds (see Table 1 and
Fig. 2). At the same time, if soil pore waters are affected by the presence of minerals, then it should impact primarily the
lithogenic elements (Ca, Mg, Sr, Si, Ti, Al, Zr…) whose concentration should be higher in negative forms of relief
relative to that in the positive ones. This hypothesis is not supported by the concentration pattern of inorganic

constituents of porewaters as shown in the next section. Note also that, because the mounds thaw later than hollows, the period of unfrozen exchange of constituents in the soil with porewater is shorter in mounds compared to hollows. However, this does not go in line with the observed difference of higher DOC and metal concentration in porewater of mounds relative to hollows.

The 2[nd] explanation of the elevated DOC concentration in mounds compared to hollows across the whole permafrost gradient is related to the time of reaction between the peat and the pore fluids. From detailed hydrological studies on frozen peatbog of western Siberia, the water residence time in peat mound is a factor of 14 higher than that in hollows and depressions (Novikov et al., 2009). The latter have much higher hydrological connectivity to surrounding streams and temporary water channels and as such offer shorter contact time and pathways of vertically infiltrating and laterally migrating water. During the summer baseflow period, up to 70-80% of watershed covered by mounds in frozen peatland of western Siberia may remain disconnected from the hydrological network (Batuev, 2012). The mounds and polygons are therefore essentially controlled by water evaporation, leading to evaporative concentration of DOC and other solutes within the soil profile. The available data on water infiltration parameters of hollows and permafrost subsidences located in discontinuous permafrost zone of the WSL demonstrate an order of magnitude faster water migration in various depressions (hollows, subsidences) compared to mounds (Novikov et al., 2009 and unpublished data of the authors on NaCl tracer migration in frozen polygons and palsa peatbogs of the WSL). The density of the peat in the mounds and polygons is a factor of 2 to 10 higher than that in the hollows and depressions (Ivanov and Novikov, 1976). Thus an analogy of ground surface and deep peat can be used for comparison between negative and positive forms of microrelief, respectively. In the peatland-dominated zone of discontinuous permafrost, the total porosity was reported to drop by about 10% between the ground surface and 35 cm depth; however, the active porosity decreased by as much as 40% over the same distance (Quinton et al., 2000). The saturated hydraulic conductivity of peat decreases rapidly with depth (Quinton et al., 2009). It thus can be hypothesized that, in the dense peat on mounds and polygons, the pores are significantly smaller with less interconnection, which leads to more restricted flow and greater turtuosity (Rezanezhad et al., 2009, 2010, 2016). All these factors should increase the water residence time in pores of peat in mounds relative to hollows and allow for efficient enrichment of peat porewater by DOC in the former.

The DOC pore water concentration invariance across the latitudinal gradient of the WSL is consistent with the lack of peat thickness and thermal regime effect on pore water chemistry. First, the peat thickness did not exert a direct impact on the degree of porewater enrichment in DOC among various micro-landscapes: there was no dependence between the DOC concentration in porewater and the total thickness of the peat ($R^2 < 0.01$, not shown). Second, the thermal regime of soil porewater is responsible neither for the difference between mounds and hollows nor for latitudinal dependence of DOC concentration. The effect of temperature on peat leaching in aqueous solution is not known, but by analogy with surface-controlled dissolution reaction of minerals (i.e., Schott et al., 2009) it can be by a factor of 2 to 3 for each 10°C rise. Such a large difference in 10°C between different adjacent micro-landscape seems highly unlikely. This is confirmed by both our field measurements in Tazovsky (mean annual temperature of peat at 5 cm depth is equal to -1.9°C in mound and +1.9°C in hollow), and the observations of other researchers in the WSL. In the Nadym region (discontinuous permafrost zone), the mean annual temperature of mounds and hollows is 1.0 and 1.6°C, respectively (Bobrik et al., 2015). At the latitude of Urengoy-Tazovsky and Khanymey, the average difference between mound and hollow of summer-time temperature at 20 cm depth is 2.9 and 3.4°C, respectively (Novikov et al., 2009). A similar difference of peat temperature between mounds and depressions at 20 cm depth (< 4°C) was reported for the Noyabrsk region (discontinuous permafrost zone, Makhatkov and Ermolov, 2015). Globally, the temperature of soil porewater

across the latitudinal gradient does not exceed 10°C (Novikov et al., 2009) which is not sufficient to exert any
pronounced control on DOC concentrations.
To summarize, we hypothesize that *i*) the DOC concentration should be controlled by the DOC residence time
and travel pathway through the organic topsoil and *ii*) the enrichment in DOM of the interstitial soil solution occurs via
lichens, moss, litter and peat leaching. Although the runoff is known to exert the primary control on stream DOC export
from the boreal peatland catchments (Olefeldt et al., 2013; Leach et al., 2016), the existing hydrological modeling of
subsurface transport of dissolved carbon in a discontinuous permafrost zone suggests that both concentration and load of
DOC are water flow-independent (Jantze et al., 2013). As such, it is the time of reaction between the peat and downward
infiltrating waters that essentially controls the degree of peat pore-water enrichments in DOC. This time is presumably
similar across significant permafrost and climate gradients.

### 4.2. Factors controlling major and trace element concentration in peat soil porewaters

Organic and organo-Fe, Al colloids dominate the speciation of most cations (including alkaline-earth metals)
and TE in low-TDS humic surface waters of permafrost-affected WSL territory (Pokrovsky et al., 2016b), similar to
other boreal catchments (Köhler et al., 2014). As a result, the behaviour of many major and TE in peat porewater is likely
to follow that of DOC, Fe and Al as main colloidal carriers. The importance of colloidal Fe and Al as primary carriers of
TE in peat soils is confirmed by results of this study: in pore-waters, none of the TE correlated with DOC ($R < 0.5$)
whereas Fe and Al concentrations correlated with many TE such as Ti, V, Cr, Co, Ni, As, Sr, Zr, Nb, heavy REE, Hf.
This is consistent with decoupling of $TE^{3+}$ and DOC during size separation procedure as two independent colloidal pools
(high molecular weight Fe, Al-rich and low molecular weight $C_{org}$-rich), already demonstrated for European boreal rivers
(Neubauer et al., 2013; Vasyukova et al., 2010) and other Siberian rivers and WSL thermokarst lakes (Pokrovsky et al.,
2006; Pokrovsky et al., 2011, 2016b). At the same time, although organo-ferric and organo-aluminium colloids are
certainly important factors of insoluble element transport in peat soil, the source of TE may become more limiting for
overall concentration of TE in soil porewater than their speciation. There are two possible sources of "lithophile"
elements in the peat and peat porewaters: atmospheric dust deposition at the moss and lichen surface and upward
migration of soil fluids that carry mineral particles from underlying loam horizons. The loam horizons are rich in silicate
clay minerals (e.g., Ovchinnikov et al., 1973; Golovleva et al., 2017) that contain insoluble elements. The geochemical
analysis of TE distribution in WSL peat cores across the studied permafrost gradient allowed to distinguish several
categories of TE depending on their source such as soluble atmospheric aerosols, atmospheric dust, underlying mineral
layers, plant biomass, and surface water flooding (Stepanova et al., 2015). The atmospheric deposition of lithogenic
elements in the form of soluble aerosols on the moss surfaces followed by incorporation into the peat is expected to be
low as shown by thorough snow analyses across the large WSL gradient (Shevchenko et al., 2016). Therefore,
atmospheric dust seems to be the main source of insoluble metals in WSL peat as it is also known from other northern
bogs (Shotyk et al., 2016). Regardless of the origin of lithophile elements, we hypothesize that the leaching of insoluble
trivalent and tetravalent hydrolysates ($TE^{3+}$, $TE^{4+}$) from solid phase to interstitial soil solution may be restricted by the
availability of silicate clay minerals within the peat core.
Based on results of the PCA treatment (**Fig. S3 A, B**), the dendrogram of a hierarchical cluster (**Fig. S3 C**) and
the correlations between elements (**Table S1**) we hypothesize that the source of Cr, V, Al, REEs, Nb, Zr, Hf, Th, U but
also of Mg and Li is silicate minerals dispersed within the peat matrix. These elements exhibit the highest correlation
with Si in porewaters and appear to be linked to the first factor (F1) of the PCA. The silicate minerals may originate from
both atmospheric dust and underlying clay/silt horizons. The lack of correlation of K, Rb, Mn, Ba, Mo, W, Zn, Pb, Cd,
Cs, Sb with DOC, Fe or Al in peat porewaters of WSL (**Table S1**) can be explained by specificity of these elements. In
particular, K, Rb, Mn, Cu, Ba are biotically-controlled by moss growth and thus unlikely to be linked to any mineral
source (Stepanova et al., 2015). It seems also plausible that indifferent oxyanions (Mo, Sb, W) or disperse pollutants
delivered by atmospheric deposition on moss surface followed by incorporation into peat (Zn, Cd, Pb, Sb, Tl) do not
exhibit significant correlation with main colloidal components.
One can expect that dissolved element decreases its concentration in the peat porewater northward regardless of
the micro-landscape due to *i*) decrease of the thickness of peat deposits in total and the active soil (peat) layer in
particular (Beilman et al., 2009; Novikov et al., 2009: Stepanova et al., 2015) which decreases the amount of peat
interacting with downward penetrating fluids; *ii*) decrease of plant biomass (Frey and Smith, 2007), which diminishes the
amount of plant litter that can release the elements (Pokrovsky et al., 2006; Fraysse et al., 2010), and also decrease the
plant ability to weather minerals within the soil profile (Moulton et al., 2000); *iii*) shortening the unfrozen period of the
year leading to the decrease of the residence time of water in soil pores and *iv*) overall decrease of the intensity of
chemical weathering, $CO_2$ consumption and riverine fluxes with mean annual temperature decrease (Dessert et al., 2003).
However, an unexpected result of this study was that the overwhelming number of major and TE did not exhibit any
statistically significant decreasing trend of concentration with latitude. Instead, we observed a measurable northward
increase in concentration of a number of lithogenic elements, whose presence is known to mark the intensity of mineral
weathering. These are Mg, Al, Ti, V, Sr, REEs, Zr, Hf and Th, originated from silicate minerals of the soil profile. For
example, Al, Ba, Fe, and Mn were reported to reflect the mineral weathering as they exhibited elevated concentrations in
Alaskan rivers during the late Fall, that correlated with the maximal depth of the thawed active layer (Barker et al.,
2014). The mechanism related to enhanced mobilization of low-soluble elements during deepening of the ALT is
penetration of DOM-rich surface fluids to deeper soil horizon and leaching of lithogenic elements from underlying
mineral substances, in the form of strong organic complexes (chelates). This mechanism can be tested via comparison of
lithogenic element concentration in contrasting micro-landscapes. Thus, Sr, which is considered as an indicator of
mineral sources in surface waters of the permafrost zone (Keller et al., 2010; Bagard et al., 2011), was highly similar
between mound and hollow or even higher in mounds than in hollows or subsidences (Table 2). Given that the negative
forms of relief in the WSL exhibit higher proximity of thawed layer to the mineral horizon because of lower thickness of
peat and deeper ALT (Tyrtikov, 1973; Lupachev et al., 2016), the lack of link between Sr concentration and ALT
position within the peat-silt/clay profile suggests that the underlying minerals do not participate in feeding the soil
solutions by lithogenic elements. Rather, aeolian (long-range) dust deposits throughout the territory may lead to
incorporation of solid atmospheric particles into the moss biomass. Subsequently, it is the dissolution of agglutinated
minerals that enriches the peat porewater in lithogenic elements, including Si. Moreover, the concentration of elements
likely originated from silicate matrix (Al, Si, Fe) in hollows and subsidences did not exceed that in mounds. Taken into
account that the position of the permafrost boundary is much closer to the mineral substrate in negative forms of relief
compared to mounds (see Table 1 and Fig. 2), this strongly suggests the lack of element leaching from the underlain
mineral matrix. As such, the observed trends of element concentration with latitude reflect the leaching of essentially peat
constituents with associated silicate particles without interferences with massive deposits of underlying sand, clay and silt
in various micro-landscapes. Following the same reasoning, the lack of DIC, Mg and Ca variation among the micro
landscapes suggests a negligible role of silicate and carbonate mineral weathering within the peat profile.

In addition to evaporative concentration mechanism and the greater residence time of solutes in mound

compared to hollows, identified for DOC pattern in section 4.1, the peat chemical composition may be different between
negative and positive forms of relief and thus it can contribute to porewater enrichment in major and TE. Indeed, the
degree of peat decomposition and elementary content of peat on mounds is higher than that on hollows and depressions
(Stepanova et al., 2015): a comparison of peat elementary composition at 15 cm depth on Pangody site demonstrated a
factor of 1.5 to 3.5 higher concentration in mounds compared to hollows of major (Ca, K, Na, Fe) and ~40 TE except
Mg, Zn, Sb and Pb (a factor of 1.3 to 3 richer in hollows than in mounds).

The lack of increase of Cl, $SO_4$ and Na in peat porewaters from the most northern site (Tazovskiy) compared to

the intermediate sites (Urengoy, Pandogy) dismisses the possibility of element leaching from frozen saline sediments
abundant in the Russian Arctic Coast (e.g., Brouchkov, 2002). Presumably, these saline sediments are not in contact with
soil and suprapermafrost waters even at the time of maximal ALT, as also inferred from riverwater geochemistry in the
permafrost-affected region of WSL (Pokrovsky et al., 2015). The elements originated from marine aerosols such as Na,
Cl, $SO_4$, B, Li, Rb, Cs exhibited a decreasing or indifferent, but not increasing trend of concentration northward. This
precludes a strong influence of marine atmospheric deposition on surface water chemistry, unlike it was suggested in
earlier works in this region (Syso, 2007; Smolyakov, 2000).


*4.3. Comparison of peat porewaters with rivers and thermokarst lakes*

The peat soil porewaters sampled above the position of the permafrost table can serve as representative sources

of water and solutes prior to export to the thermokarst lakes and rivers (Fig. 2). Therefore, a first-order comparison of
concentrations between these aquatic systems allows evaluation of the role of peat (shallow surface) versus mineral (deep
subsurface and underground waters) feeding of Siberian inland waters. This comparison was based on mean values of
DOC and TE concentration in porewaters for the whole permafrost-affected WSL territory (Table 2) and those previously
published for lakes and rivers of the same latitudinal gradient (Manasypov et al., 2014 and Pokrovsky et al., 2015,
2016a). The dissolved components measured in rivers and lakes during summer period can be classified into three
categories: (1) Rivers or lakes exceed soil porewaters by a factor of 3 to 10; (2) River or lakes are similar to porewaters
within a factor of 2, and (3) Rivers or lakes are significantly lower (more than a factor of 3) than the porewaters. The
elements of the first category are DIC, Ca, Mg, Si, B, Al, Mn, Na for rivers and only Si for lakes. The second category
comprises DOC, Li, K, Rb, Fe, Ni, Co, Cr, As, Sr and U for rivers and Li, B, Na, K, Rb, Cs, Ca, Mg, Ti, V, Mn, Ni, Cu,
Zn, Co, Cd, Sr, Mo, As, Sb for lakes. The 3[rd] category includes Ti, Cu, Pb, Cd, Mo and REEs for rivers and DOC, Al, Fe,
Ga, Y, Zr, Ba, W, REEs, Th, U for lakes. This first-order comparison demonstrates that the soil porewaters alone are
sufficient to provide the concentrations of all major and TE in lakes. In other words, the transport of soil porewaters
along the permafrost boundary in the form of suprapermafrost flow may be the sole source of incoming solutes to
thermokarst lakes of western Siberia, across all 3 permafrost zones. This hypothesis is fully consistent with the lack of
any underground feeding of WSL thermokarst lakes, demonstrated in earlier studies (Manasypov et al., 2015).

In contrast to lakes that can be fully supplied by solutes from surrounding peat porewaters, the rivers require

some "mineral" influx in addition to surface and shallow subsurface "organic" influx, in order to explain the elevated
concentrations of DIC, Ca, Mg, Na, Si, Al in the riverwater relative to the peat porewater. This influx, mostly
pronounced during summer baseflow period, may include the groundwater seeping via taliks on the river bed and shallow
subsurface flow over clays and silt deposits. This process is fairly well known for other, non-peatland permafrost setting
(MacLean et al., 1999; Bagard et al., 2011; Barker et al., 2014; Tank et al., 2016).
The latitudinal dependences of element concentration in the peat pore water revealed in this study can be
compared to the latitudinal dependences of DOC and element concentration in adjacent thermokarst lakes and rivers. The
elementary trends in the inland waters of western Siberia were associated to the influence of marine aerosols or long-
range atmospheric transport of industrial pollutants in lakes (Manasypov et al., 2014) and the evolution of chemical
composition of the peat and underlying mineral deposits in rivers (Pokrovsky et al., 2015; 2016a). However, the possible
links are not straightforward and valid only for a small number of elements. Thus, increasing concentrations of Ca, Ni
and Sr (**Fig. 4C, 4G, 4H**, respectively) and decreasing concentration of Sb and Pb (**Fig. 5 D and E,** respectively)
northward are consistent with the trend in thermokarst lakes of western Siberia from 63°N to 71°N (Manasypov et al.,
2014). However, the other elements exhibiting a clear increasing (K, Cu, Mo) or decreasing (V, Ba) latitudinal trend in
lakes (Manasypov et al., 2014) do not show such a trend in peat pore-waters sampled in this study. Presumably, variable
and simultaneously acting processes control the delivery of element from the peat core to the adjacent lakes over the
permafrost gradient.
Because the leaching of peat constituents by downward penetrating fluids is very fast and weakly depends on
temperature and local hydrological pathway within the peat pores, one can expect that the global hydrological setting will
primarily control the peat weathering intensity. As such, it is the amount of water that passes through the peat soil
column before being evacuated to the river that defines the overall export fluxes of elements from the peatland to the
hydrological network. This prediction is consistent with reported higher riverine fluxes of DOC, Si and cations in the
northern region of the WSL (66.5 to 67.5°N) relative to the southern region (62-65°N) of this territory corresponding to
higher surface runoff in the north (Pokrovsky et al., 2015).
The fluxes of Ca, Mg and $HCO_3^-$ ions carried by rivers are used for calculation the $CO_2$ uptake flux due to
chemical weathering, i.e., reaction of atmospheric $CO_2$ with Ca, Mg-bearing silicate minerals (Dessert et al., 2003;
Beaulieu et al., 2012). Not more than 10% of total riverine flux of Ca, Mg and $HCO_3$ is considered to be due to
atmospheric input. An important consequence of our obtained results on soil porewaters in the WSL is that the intensity
of chemical weathering and associated $CO_2$ consumption in the permafrost regions (i.e., Beaulieu et al., 2012) by small
rivers without pronounced underground feeding in peatlands could be overestimated relative to the regions with shallow
organic soil horizons. As a result, the flux of DIC and major cations in the peatland-draining rivers should be corrected
for the input of these elements via peat pore-water discharge to the river main stream. For a number of small rivers
($S_{watershed}$ < 1000 km²) in the permafrost zone of the WSL that are fed by shallow surface runoff through the peat horizon,
this correction can range from 20 to 80% of total riverine DIC, Ca and Mg flux. The global consequence of this
correction is that the continental-weathering $CO_2$ sink in northern peatland regions might be a factor of 2 to 4 smaller
than that currently deduced from the fluxes of large rivers.


*4.4. Prospective for climate change in western Siberia*
In accordance with a common scenario of the climate change in the subarctic, a shift of the permafrost boundary
further north and the increase of the active layer thickness are anticipated in the WSL (Pavlov and Moskalenko, 2002;
Frey and McClelland, 2009; Moskalenko, 2009; Romanovsky et al., 2010; Vasiliev et al., 2011; Anisimov et al., 2013).
This agrees with large-scale permafrost shifts consisting in southern boundaries moving northward (see Walvoord and
Kurylyk, 2016 for a review). Assuming a "substitution space for time" scenario, and upscaling the data of peat pore
waters obtained in this study, we predict that the shift of the permafrost boundary northward even by 2° latitude will not
affect the concentrations of most major and TE in peat pore-waters. The concentrations of DOC, DIC, Ca, Mg, K, Al, Fe,
and trace metals in continuous permafrost zone may remain constant or decrease by a factor of 1.5 to 2 which is often
within the natural variation between different microlandscapes, soil depths and seasons.
The ALT is projected to deepen more than 30% during this century in the Northern Hemisphere (Anisimov et
al., 2002; Stendel and Christensen, 2002; Dankers et al., 2011). As a general scenario in frozen peatlands of the subarctic,
this increase will bring about the involvement of mineral horizons into water infiltration zone downward the soil profile
(Walvoord and Kurylyk, 2016). The degradation of peat mounds and polygons will be accompanied by the spreading of
hollows and depressions (Pastukhov and Kaverin, 2016). As a result, the water coverage of the watershed will increase
thus enhancing the anaerobic conditions. On the one hand, this will increase the fraction of hollows and depressions
containing less concentrated interstitial soil solutions and thus the stock of DOC, major elements and trace metals in soil
fluids will decrease. On the other hand, the increasing anaerobic conditions may preferentially mobilize redox sensitive
elements (Fe, Mn, Cr, V…) from the peat to the porewaters. Overall, the share of spring runoff from the mounds to the
rivers and lakes will decrease whereas during the summer baseflow, the input from the hollows and depressions to the
hydrological network will increase.
The concept "substituting space for time" allows foreseeing the consequences of soil warming in the continuous
permafrost zone of the WSL peatlands on the adjacent river chemistry and export of carbon and metals from the
watersheds. This prediction can be made only for small rivers of the WSL (e.g., watershed area < 10,000 km²) which
drain the adjacent peatlands, have no underground feeding and flow essentially during unfrozen period of the year (see
Pokrovsky et al., 2015, 2016a). For this, two basic scenarios can be considered: (*i*) a constant latitudinal pattern of
permafrost distribution (no boundary migration) but complete disappearance of peat mounds and their replacement by
hollows and depressions and (*ii*) a shift of the permafrost boundary to the north and transformation of the continuous
permafrost zone into the discontinuous and transformation of the discontinuous permafrost into the sporadic without
changing the microlandscape distribution. As a first approximation, we assume no change in precipitation,
evapotranspiration and riverine runoff in the northern part of WSL (60-68°N), given that the drying trend will be
pronounced only in the regions located to the south of 60 °N (Alexandrov et al., 2016).
The first scenario yields a decrease in the concentrations of DOC, DIC, major cations and trace metals in
porewaters of continuous permafrost zone by not more than 30%. This estimation stems from the maximal difference in
element concentration between mounds and hollows (Table 2) and typical proportion of mounds in the terrestrial
landscape of the WSL (35±15 %, Novikov et al., 2009 and authors' unpublished data). The second scenario is based on
the latitudinal patterns of element concentration in the peat porewaters (Table 3 and Figs 3-5, S4, S5). For this, a linear
dependence of element concentration in all microlandscapes on the latitude given in Figs. 3 to 5 can be used. A two-
degree northward shift in the position of the permafrost boundaries will bring about a factor of 1.3±0.2 decrease in Ca,
Mg, Sr, Al, Fe, Ti, Mn, Ni, Co, V, Zr, Hf, Th, and REEs concentration in continuous and discontinuous permafrost
zones. Note that a possible decrease in DOC, $SUVA_{280}$, Ca, Mg, Fe, Sr will not exceed 20% of their actual values.
Finally, there may be an increase in Cl, Na, K, Rb, Cs, Zn, P and Sb concentration by 30±10%. In both scenarios of
permafrost thawing in the WSL peatlands we do not expect any sizeable increase of soil porewater concentration in DOC
and metal and enhancement of the export of solutes by small-size rivers which are not connected to the underground
reservoirs. This contradicts the dominating paradigm of the increase of DOC, DIC, major cations and metal discharge
from the land to the ocean upon the on-going climate warming in other permafrost regions. Combining both scenario of
permafrost thaw (northward permafrost boundary shift and extending the hollows over mounds) suggests that over the
first decades, relatively fast permafrost coverage shift will not be accompanied by the change of micro-landscapes and
thus the overall decrease of DOC and metal concentration in peat porewaters will be around 20 to 30%. The average rate
of peat formation in Siberian flat-mound bogs is 0.24 mm $y^{-1}$ (Inisheva et al., 2013). Thus, taking into account the climate
warming and accelerated peat growth, after 500 to 1000 years which are necessary to form the new ca. 20-cm peat layer,
the second scenario will take over and thus up to 2-fold cumulative element concentration decrease in soil fluids of
continuous permafrost zone may occur. Assuming a dominant feeding of small rivers by soil porewaters transported
along the permafrost boundary, a slight decrease (i.e., < 30 %) of riverine transport of DOC, DIC, Fe, Al, Ca, Mg from
the northern part of the WSL territory to the Arctic Ocean is anticipated. This decrease will be mostly pronounced for
small rivers such as those of the Arctic coastal zone.

## Conclusions

A snopshot of peat soil water chemistry allowed to quantify the distribution of DOC, major and trace element in
peat porewaters at the end of the active period across a sizeable gradient of permafrost. We did not confirm a trend of
diminishing DOC and metal concentration in peat porewaters northward, despite a decrease in mean annual temperature,
vegetation density and the active layer thickness. DOC, DIC and most major and TE did not exhibit any statistically
significant trend of concentration with the latitude. A clear trend of increasing concentrations of Mg, Ca, Al, Ti, V, Ni,
Sr, heavy REE, Zr, Hf and Th marked the increase of the influence of silicate mineral weathering. Concentrations of
DOC, $SO_4^{2-}$, B, V, Cs, Th in pore waters in the peat mounds usually exceeded those in hollows and permafrost
subsidences. The water residence time in peat of various densities and the peat chemical composition werehypothesized
to be the main factors controlling the degree of element leaching from the peat column to the pore fluids. Applying a
"substituting space for time" approach for the climate warming scenario in the WSL, we predict that the northward
migration of permafrost boundary and the replacement of thawing frozen peat mounds and polygons by hollows,
depressions and subsidences will decrease the concentrations of DOC, DIC, major cations and trace metals in porewater
of continuous permafrost zone by a factor of 1.3±0.2. This in turn will decrease the feeding of small rivers and lakes by
peat soil leachates and the overall export of DOC and metals from the WSL territory to the Arctic Ocean may decrease.
As such, the dominating paradigm of the increase of DOC, DIC, major cation and metal export fluxes upon the on-going
climate warming in boreal and subarctic regions should be revised for the case of frozen peatlands.


**Data availability**

Full data set of major and trace element concentration in porewaters (< 0.45 μm) across the latitudinal profile of Western
Siberia Lowland is available at the Research Gate,
https://www.researchgate.net/publication/313058330_Element_concentrations_in_peat_soil_solutions_across_the_micro
-landscapes_and_permafrost_zones_of_western_Siberia_peatlands


**Acknowledgements**

We acknowledge support from RFBR Nos. 16-34-60203 mol_a_dk, BIO-GEO-CLIM grant from the Russian Ministry of
Science and Education and Tomsk State University (No 14.B25.31.0001), RFFI grants No 15-29-02599, 17-55-16008,

FCP "Kolmogorov" Minobrnauki RF RFMEFI58717X0036, and a partial support from and RSF (RNF) grant No 15-17-10009 "Evolution of thermokarst ecosystems".

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

Table 1. Physico-geographical, permafrost and soil parameters of 5 study sites.

| Site | Latitude, °N | MAT, °C | Mean annual precipitation, mm | Mineral substrate | Micro-landscapes | Peat thickness, m | Seasonal thaw depth, cm | Soil type (WRB, 2014) |
|---|---|---|---|---|---|---|---|---|
| Tazovsky, (Tz) | 67.4 | −9.1°C | 363 | Clay loam and loam | polygon | 2.0–4.0 | 41 | Dystric Hemic Epicryic Histosols (Hyperorganic); Dystric Murshic Hemic Epicryic Histosols (Hyperorganic) |
| | | | | | permafrost subsidences | | 55 | Dystric Epifibric Hemic Cryic Histosols (Hyperorganic) |
| | | | | | frost crack | | 44 | Dystric Epifibric Cryic Histosols (Hyperorganic) |
| | | | | | hollows | 0.2–1.5 | 65 | Dystric Fibric Cryic Histosols; Histic Reductaquic Cryosols (Clayic) |
| Urengoy, (Ur) | 66.1 | −7.8°C | 453 | Loam and silt loam | peat mounds | 2.0–2.5 | 49 | Dystric Hemic Epicryic Histosols (Hyperorganic) |
| | | | | | hollows | 0.3–1.2 | 98 | Histic Reductaquic Cryosols (Loamic); Dystric Fibric Histosols (Gelic) |
| Pangody, (Pg) | 65.9 | −6.4°C | 484 | Loam | peat mounds | 0.2–1.3 | 49 | Dystric Hemic Epicryic Histosols; Histic Cryosols (Loamic); Histic Oxyaquic Turbic Cryosols (Loamic) |
| | | | | | permafrost subsidences | 0.6–1.1 | 74 | Dystric Hemic Endocryic Histosols |
| | | | | | hollows | 0.3–1.0 | 82 | Dystric Epifibric Endocryic Histosols; Histic Reductaquic Turbic Cryosols (Loamic); Dystric Fibric Histosols (Gelic) |
| Khanymey, (Kh) | 63.8 | −5.6°C | 540 | Sand | peat mounds | 0.1–1.4 | 90 | Dystric Hemic Cryic Histosols; Spodic Histic Turbic Cryosols (Albic, Arenic); Histic Turbic Cryosols (Albic, Arenic) |
| | | | | | permafrost subsidences | 0.7–1.1 | 165 | Dystric Hemic Histosols (Gelic) |
| | | | | | hollows | 0.4–1.1 | 215 | Dystric Epifibric Histosols; Spodic Histic Turbic Cryosols (Arenic); Gleyic Histic Entic Podzols (Turbic) |
| Kogalum, (Kg) | 62.3 | −4.0°C | 594 | Sand | ridge | 1.7–2.3 | – | Dystric Ombric Fibric Histosols (Hyperorganic) |
| | | | | | hollows | 1.0–1.5 | – | Dystric Ombric Fibric Histosols |






Table 2. Mean values of DOC, major and TE concentration with S.D. of elements in various microlandscape across the permafrostgradient. Concentrations of DOC, DIC, Cl⁻,
$SO_4^{2-}$, Ca, Mg, K, Al, Fe, Si, and Na are given in ppm and all other trace elements are in ppb.

| Elements | Kogalym (62.259°N) | | Khanymey (63.785°N) | | | Pangody (65.873°N) | | Urengoy (66.085°N) | | | Tazovsky (67.367°N) | | | WSL-mean mound/ poligon | WSL mean hollow |
|---|---|---|---|---|---|---|---|---|---|---|---|---|---|---|---|
| | mound n=4 | hollow n=2 | mound n=20 | hollow n=4 | subsidence n=4 | mound n=8 | hollow n=4 | mound n=3 | hollow n=4 | subsidence n=2 | poligon n=12 | hollow n=7 | frost crack n=4 | | |
| DOC | 50.56±15.6 | 33.7±4.1 | 82.9±29.7 | 49.6±13.5 | 76.5±21 | 90.2±55.3 | 81.58±15 | 74.28±25.2 | 50.2±3.64 | 97.9±19.9 | 72.9±12.9 | 52.53±7.7 | 58.4±30.8 | 79.8 | 58.1 |
| DIC | 1.45±0.27 | 1.42±0.3 | 1.65±0.36 | 1.42±0.05 | 1.7±0.11 | 1.84±0.35 | 1.54±0.46 | 1.36±0.17 | 1.58±0.7 | 1.32±0.17 | 1.44±0.18 | 1.68±0.13 | 1.76±0.42 | 1.59 | 1.56 |
| Cl⁻ | 0.61±0.5 | 0.91±0.06 | 0.49±0.4 | 0.26±0.17 | 0.31±0.16 | 0.52±0.43 | 0.68±0.45 | 0.47±0.33 | 0.54±0.41 | 0.53±0.21 | 0.20±0.18 | 0.18±0.09 | 0.28±0.15 | 0.42 | 0.43 |
| $SO_4^{2-}$ | 0.13±0.03 | 0.16±0.09 | 0.64±0.47 | 0.15±0.02 | 0.14±0.06 | 0.41±0.35 | 0.24±0.18 | 0.81±0.14 | 0.16±0.05 | 0.17±0.03 | 0.60±0.44 | 0.067±0.04 | 0.13±0.10 | 0.56 | 0.14 |
| Ca | 1.03±0.34 | 1.07±0.57 | 0.74±0.52 | 1.34±0.17 | 0.97±0.14 | 1.33±0.4 | 1.14±0.16 | 1.13±0.22 | 1.17±0.35 | 0.97±0.17 | 2.04±1.7 | 1.78±1.03 | 1.8±0.4 | 1.31 | 1.40 |
| Mg | 0.13±0.07 | 0.12±0.05 | 0.14±0.11 | 0.21±0.09 | 0.13±0.04 | 0.28±0.22 | 0.35±0.27 | 0.12±0.03 | 0.19±0.18 | 0.07±0.001 | 0.3±0.29 | 0.34±0.3 | 0.36±0.16 | 0.20 | 0.27 |
| K | 1.06±0.49 | 1.16±0.26 | 0.32±0.13 | 0.34±0.26 | 0.31±0.06 | 0.99±0.62 | 0.79±0.33 | 0.21±0.06 | 0.16±0.05 | 0.18±0.004 | 0.26±0.17 | 0.19±0.06 | 0.14±0.1 | 0.47 | 0.42 |
| Al | 0.13±0.06 | 0.15±0.03 | 0.19±0.12 | 0.26±0.04 | 0.20±0.05 | 0.39±0.26 | 0.67±0.33 | 0.31±0.15 | 0.18±0.05 | 0.17±0.03 | 0.41±0.3 | 0.37±0.22 | 0.42±0.22 | 0.28 | 0.35 |
| Fe | 1.17±1.04 | 0.96±0.6 | 0.54±0.42 | 0.76±0.21 | 0.85±0.19 | 1.97±1.05 | 1.99±1.23 | 0.90±0.04 | 1.54±0.6 | 0.87±0.13 | 1±0.73 | 1.14±0.65 | 2.19±0.97 | 0.99 | 1.28 |
| Si | 1.94±1.45 | 1.12±0.33 | 1.04±1.27 | 0.6±0.18 | 0.82±0.32 | 2.94±1.44 | 3.08±1.7 | 0.49±0.14 | 0.82±0.38 | 0.38±0.03 | 1.12±0.97 | 1.27±1.35 | 1.77±1.51 | 1.39 | 1.42 |
| Li | 0.46±0.04 | 0.63±0.10 | 0.45±0.42 | 0.39±0.05 | 0.40±0.20 | 1.14±0.76 | 1.11±0.63 | 0.17±0.01 | 0.37±0.36 | 0.17±0.01 | 0.36±0.15 | 0.80±0.71 | 0.44±0.25 | 0.53 | 0.68 |
| B | 1.39±0.57 | 3.39±0.07 | 4.09±2.02 | 2.97±0.97 | 2.91±1.99 | 2.19±1.29 | 2.03±1.16 | 0.63±0.34 | N.D. | N.D. | 3.54±1.52 | 1.31±0.71 | 2.38±0.85 | 3.26 | 2.13 |
| Na | 0.44±0.25 | 0.45±0.09 | 0.28±0.12 | 0.35±0.15 | 0.26±0.03 | 0.39±0.2 | 0.50±0.11 | 0.23±0.1 | 0.25±0.22 | 0.14±0.02 | 0.19±0.08 | 0.26±0.10 | 0.20±0.1 | 0.29 | 0.34 |
| Ti | 2.33±1.21 | 0.66±0.21 | 2.92±2.02 | 2.02±0.48 | 3.23±0.8 | 3.8±1.57 | 3.68±1.58 | 1.72±0.36 | 1.38±0.32 | 1.43±0.02 | 3.69±0,71 | 3.48±1.34 | 5.25±2.78 | 3.07 | 2.54 |
| V | 0.51±0.38 | 0.28±0.18 | 0.43±0.26 | 0.35±0.22 | 0.56±0.114 | 0.67±0.22 | 0.96±0.67 | 0.77±0.47 | 0.26±0.082 | 0.28±0.09 | 1.71±1.51 | 0.97±0.52 | 1.63±0.99 | 0.83 | 0.65 |
| Cr | 0.54±0.28 | 0.31±0.11 | 1.12±0.36 | 1.17±0.56 | 1.23±0.42 | 1.12±0.4 | 1.34±0.39 | 0.27±0.18 | 0.39±0.2 | 0.203±0.001 | 0.93±0.38 | 0.86±0.31 | 1.22±0.65 | 0.97 | 0.87 |
| Mn | 6.89±3.3 | 10.8±0.4 | 3.33±2.95 | 3.05±1.6 | 2.64±1.34 | 11.3±8.5 | 5.77±4.25 | 6.05±2.02 | 14.38±5.54 | 9.31±1.58 | 58.9±37.3 | 47.3±40.0 | 59.1±34.33 | 19.7 | 21.21 |
| Co | 0.18±0.04 | 0.16±0.12 | 0.22±0.11 | 0.29±0.1 | 0.34±0.09 | 1.18±0.54 | 1.24±0.65 | 0.26±0.09 | 0.34±0.14 | 0.21±0.03 | 0.99±0.63 | 0.92±0.62 | 1.43±0.46 | 0.59 | 0.677 |
| Ga | 0.05±0.04 | 0.02±0.01 | 0.51±0.45 | 0.06±0.02 | 0.55±0.44 | 0.07±0.03 | 0.15±0.15 | 0.59±0.22 | 0.42±0.18 | 0.32±0.01 | 0.20±0.18 | 0.31±0.23 | 0.51±0.42 | 0.32 | 0.224 |
| As | 1.00±0.49 | 0.76±0.2 | 0.53±0.31 | 0.96±0.3 | 0.74±0.32 | 0.83±0.6 | 1.07±0.86 | 0.2±0.06 | 0.17±0.06 | 0.105±0.075 | 1.12±0.98 | 0.96±0.37 | 1.90±0.89 | 0.76 | 0.796 |
| Rb | 0.93±0.53 | 0.35±0.2 | 0.48±0.36 | 0.62±0.31 | 0.47±0.46 | 0.72±0.58 | 0.33±0.17 | 0.23±0.22 | 0.27±0.15 | 0.056±0.035 | 0.37±0.28 | 0.56±0.50 | 0.53±0.26 | 0.52 | 0.454 |
| Zr | 0.10±0.10 | 0.02±0.001 | 0.21±0.23 | 0.13±0.06 | 0.24±0.15 | 0.33±0.23 | 0.56±0.3 | 0.14±0.06 | 0.19±0.2 | 0.066±0.050 | 0.54±0.45 | 0.34±0.15 | 0.53±0.24 | 0.304 | 0.281 |
| Nb | 0.01±0.005 | 0.003±0.002 | 0.013±0.009 | 0.017±0.009 | 0.011±0.003 | 0.021±0.01 | 0.026±0.016 | 0.004±0.002 | 0.004±0.001 | 0.004±0.000 | 0.018±0.012 | 0.012±0.005 | 0.02±0.01 | 0.014 | 0.013 |
| Mo | 0.037±0.02 | 0.084±0.08 | 0.09±0.07 | 0.129±0.09 | 0.11±0.01 | 0.082±0.06 | 0.075±0.036 | 0.028±0.016 | 0.028±0.008 | 0.024±0.004 | 0.064±0.021 | 0.054±0.021 | 0.12±0.08 | 0.075 | 0.070 |
| Cd | 0.19±0.035 | 0.4±0.18 | 0.34±0.54 | 0.42±0.42 | 0.56±0.5 | 0.27±0.27 | 0.13±0.04 | 0.040.019 | 0.025±0.008 | 0.008±0.004 | 0.067±0.065 | 0.04±0.027 | 0.09±0.07 | 0.223 | 0.161 |
| Ni | 1.04±0.76 | 0.55±0.24 | 0.92±0.48 | 1.51±0.62 | 1.22±0.62 | 3.29±1.26 | 3.12±1.32 | 1.43±0.7 | 1.25±0.45 | 1±0.14 | 2.9±1.95 | 2.12±0.95 | 3.53±1.54 | 1.89 | 1.859 |
| Cu | 4.44±2.7 | 2.21±0.48 | 5.36±3.74 | 1.62±0.14 | 4.27±3.46 | 5.02±3.7 | 5.78±3.95 | 6.02±4 | 5.41±2.24 | 1.82±0.23 | 5.86±3.1 | 4.05±3.05 | 2.33±0.95 | 5.39 | 4.000 |
| Zn | 9.97±6.7 | 12.48±0.5 | 7.97±4.47 | 10.16±6.4 | 10.03±6.67 | 8.14±5.4 | 3.51±0.49 | 8±5.38 | 6.34±2.04 | 1.76±0.11 | 6.34±3.32 | 7.88±3.46 | 5.77±0.36 | 7.75 | 7.626 |
| Sr | 5.37±1.05 | 4.46±3.03 | 7.62±4.42 | 8.15±2.94 | 7.87±1.08 | 10.95±2.98 | 10.7±5.35 | 5.9±2.3 | 6.5±3.6 | 4.32±0.15 | 13.1±9.02 | 8.41±3.49 | 11.7±4.22 | 9.42 | 8.312 |
| Sb | 0.06±0.04 | 0.05±0.01 | 0.05±0.03 | 0.06±0.02 | 0.042±0.016 | 0.05±0.03 | 0.037±0.011 | 0.013±0.012 | 0.013±0.004 | 0.004±0.001 | 0.032±0.01 | 0.025±0.012 | 0.032±0.01 | 0.044 | 0.034 |
| Cs | 0.032±0.03 | 0.02±0.016 | 0.036±0.028 | 0.03±0.02 | 0.04±0.03 | 0.023±0.02 | 0.018±0.01 | 0.004±0.002 | 0.006±0.004 | 0.003±0.001 | 0.012±0.013 | 0.006±0.007 | 0.056±0.03 | 0.025 | 0.015 |
| Ba | 22.5±9.3 | 18.87±9.57 | 35.7±20.6 | 33.57±22.24 | 32.5±17.7 | 22.7±13.2 | 38.8±17.7 | 18.76±6.89 | 13.83±6.35 | 10.8±0.6 | 16.77±6.85 | 16.30±5.82 | 14.99±9.11 | 26.23 | 23.64 |
| La | 0.24±0.19 | 0.15±0.04 | 0.37±0.33 | 0.25±0.17 | 0.26±0.06 | 0.348±0.208 | 0.502±0.277 | 0.354±0.26 | 0.14±0.07 | 0.112±0.05 | 0.34±0.17 | 0.23±0.10 | 0.40±0.22 | 0.346 | 0.261 |
| Ce | 0.51±0.47 | 0.22±0.11 | 0.67±0.51 | 0.53±0.44 | 0.54±0.09 | 0.725±0.484 | 1.039±0.536 | 0.66±0.53 | 0.29±0.136 | 0.236±0.1 | 0.74±0.35 | 0.51±0.21 | 0.87±0.58 | 0.685 | 0.543 |
| Pr | 0.03±0.02 | 0.015±0.014 | 0.082±0.06 | 0.059±0.057 | 0.066±0.014 | 0.08±0.06 | 0.114±0.05 | 0.05±0.034 | 0.028±0.013 | 0.022±0.01 | 0.094±0.05 | 0.06±0.032 | 0.108±0.073 | 0.079 | 0.059 |

| | | | | | | | | | | | | | | | |
|---|---|---|---|---|---|---|---|---|---|---|---|---|---|---|---|
| Nd | 0.257±0.2 | 0.088±0.04 | 0.33±0.26 | 0.26±0.21 | 0.27±0.06 | 0.34±0.22 | 0.383±0.097 | 0.194±0.13 | 0.115±0.054 | 0.086±0.037 | 0.407±0.24 | 0.24±0.13 | 0.43±0.28 | 0.338 | 0.233 |
| Sm | 0.028±0.01 | 0.01±0.0074 | 0.07±0.05 | 0.044±0.038 | 0.058±0.016 | 0.072±0.047 | 0.080±0.021 | 0.04±0.027 | 0.025±0.012 | 0.018±0.009 | 0.092±0.057 | 0.052±0.031 | 0.099±0.069 | 0.071 | 0.047 |
| Eu | 0.011±0.01 | 0.004±0.002 | 0.015±0.010 | 0.010±0.007 | 0.015±0.007 | 0.015±0.01 | 0.016±0.005 | 0.012±0.006 | 0.008±0.004 | 0.007±0.003 | 0.022±0.013 | 0.013±0.008 | 0.025±0.016 | 0.017 | 0.011 |
| Gd | 0.03±0.014 | 0.02±0.007 | 0.07±0.05 | 0.05±0.05 | 0.061±0.02 | 0.069±0.046 | 0.078±0.021 | 0.042±0.027 | 0.025±0.013 | 0.019±0.009 | 0.096±0.061 | 0.052±0.029 | 0.099±0.068 | 0.0721 | 0.049 |
| Tb | 0.007±0.006 | 0.003±0.001 | 0.014±0.004 | 0.007±0.006 | 0.009±0.003 | 0.01±0.007 | 0.012±0.004 | 0.006±0.004 | 0.003±0.002 | 0.003±0.001 | 0.014±0.01 | 0.0074±0.004 | 0.015±0.011 | 0.0123 | 0.007 |
| Dy | 0.04±0.04 | 0.017±0.002 | 0.061±0.05 | 0.041±0.034 | 0.05±0.016 | 0.055±0.037 | 0.081±0.04 | 0.031±0.02 | 0.018±0.009 | 0.016±0.009 | 0.078±0.05 | 0.0424±0.026 | 0.087±0.068 | 0.0608 | 0.042 |
| Ho | 0.008±0.007 | 0.003±0.001 | 0.011±0.008 | 0.011±0.01 | 0.009±0.003 | 0.011±0.007 | 0.012±0.003 | 0.007±0.004 | 0.004±0.002 | 0.004±0.002 | 0.016±0.011 | 0.009±0.005 | 0.018±0.014 | 0.0115 | 0.008 |
| Er | 0.021±0.019 | 0.0069±0.0057 | 0.030±0.021 | 0.023±0.022 | 0.03±0.01 | 0.031±0.021 | 0.034±0.009 | 0.017±0.01 | 0.012±0.008 | 0.009±0.004 | 0.047±0.035 | 0.0261±0.016 | 0.051±0.037 | 0.0330 | 0.022 |
| Tm | 0.0028±0.0025 | 0.0015±0.00001 | 0.005±0.004 | 0.0032±0.003 | 0.004±0.001 | 0.004±0.003 | 0.005±0.001 | 0.002±0.001 | 0.002±0.001 | 0.001±0.0004 | 0.007±0.005 | 0.0035±0.003 | 0.007±0.004 | 0.0047 | 0.003 |
| Yb | 0.0164±0.014 | 0.006±0.0047 | 0.021±0.014 | 0.018±0.018 | 0.022±0.009 | 0.026±0.017 | 0.029±0.007 | 0.014±0.008 | 0.012±0.01 | 0.007±0.004 | 0.043±0.032 | 0.0250±0.017 | 0.046±0.032 | 0.0271 | 0.020 |
| Lu | 0.0022±0.0018 | 0.0014±0.00001 | 0.0034±0.003 | 0.003±0.0025 | 0.003±0.001 | 0.004±0.002 | 0.004±0.001 | 0.002±0.001 | 0.002±0.001 | 0.001±0.0004 | 0.007±0.005 | 0.0036±0.003 | 0.006±0.004 | 0.0041 | 0.003 |
| Hf | 0.004±0.003 | 0.0013±0.0002 | 0.006±0.005 | 0.008±0.003 | 0.008±0.004 | 0.012±0.008 | 0.016±0.007 | 0.006±0.003 | 0.005±0.005 | 0.003±0.002 | 0.015±0.014 | 0.011±0.005 | 0.016±0.008 | 0.0095 | 0.009 |
| W | 0.028±0.02 | 0.01±0.0006 | 0.036±0.03 | 0.039±0.031 | 0.044±0.007 | 0.026±0.015 | 0.032±0.012 | 0.008±0.007 | 0.004±0.006 | 0.001±0.001 | 0.014±0.008 | 0.015±0.006 | 0.022±0.018 | 0.0262 | 0.020 |
| Tl | 0.011±0.008 | 0.005±0.003 | 0.007±0.004 | 0.005±0.004 | 0.007±0.002 | 0.008±0.004 | 0.009±0.007 | 0.001±0.001 | 0.002±0.001 | 0.0009±0.00. | 0.003±0.001 | 0.003±0.002 | 0.006±0.003 | 0.0059 | 0.005 |
| Pb | 1.24±0.64 | 0.59±0.06 | 1.08±0.71 | 1.03±0.47 | 0.90±0.25 | 0.70±0.32 | 0.777±0.22 | 0.49±0.42 | 0.27±0.13 | 0.13±0.0015 | 0.603±0.186 | 0.666±0.348 | 0.86±0.16 | 0.8636 | 0.674 |
| Th | 0.04±0.035 | 0.015±0.006 | 0.065±0.06 | 0.040±0.035 | 0.051±0.004 | 0.08±0.04 | 0.089±0.023 | 0.073±0.053 | 0.032±0.023 | 0.02±0.007 | 0.093±0.054 | 0.049±0.024 | 0.07±0.03 | 0.0740 | 0.049 |
| U | 0.02±0.018 | 0.014±0.008 | 0.0303±0.03 | 0.026±0.02 | 0.026±0.005 | 0.028±0.016 | 0.055±0.025 | 0.008±0.006 | 0.015±0.01 | 0.005±0.001 | 0.026±0.014 | 0.021±0.018 | 0.032±0.017 | 0.0265 | 0.026 |





 **Table 3.** Latitudinal trends of average element concentration in two main habitats persisting in all

 five study sites. L is for latitude (°N) and R² is a linear regression coefficient (Eqn. 1)

| Element | Habitat | Equation | R² |
|---|---|---|---|
| S.C. | Hollow | [S.C.] = -2.367L + 207.36 | 0.15 |
| | Mound/polygon | [S.C.] = -0.493L + 73.345 | 0.006 |
| pH | Hollow | [pH] = 0.0278L + 2.4126 | 0.035 |
| | Mound/polygon | [pH] = 0.0663L - 0.3568 | 0.515 |
| DOC | Hollow | [DOC] = 4.6937L - 251.92 | 0.31 |
| | Mound/polygon | [DOC] = 4.0364L - 188.61 | 0.29 |
| SUVA | Hollow | [SUVA] = 0.148L – 6.861 | 0.599 |
| | Mound/polygon | [SUVA] = 0.0258L + 1.192 | 0.031 |
| DIC | Hollow | [DIC] = 0.0405L - 1.131 | 0.58 |
| | Mound/polygon | [DIC] = 0.0191 L + 0.3357 | 0.1 |
| $Cl^-$ | Hollow | [Cl⁻] = -0.084L + 5.9763 | 0.33 |
| | Mound/polygon | [Cl⁻] = -0.0601L + 4.368 | 0.64 |
| $SO_4^{2-}$ | Hollow | [SO₄²⁻] = -0.0087L + 0.7179 | 0.079 |
| | Mound/polygon | [SO₄²⁻] = 0.0824L - 4.8422 | 0.41 |
| Ca | Hollow | [Ca] = 0.0612L - 2.6683 | 0.19 |
| | Mound/polygon | [Ca] = 0.1828L - 10.639 | 0.59 |
| Mg | Hollow | [Mg] = 0.0405L - 2.395 | 0.69 |
| | Mound/polygon | [Mg] = 0.0302L - 1.773 | 0.43 |
| Na | Hollow | [Na] = -0.0348L + 2.621 | 0.49 |
| | Mound/polygon | [Na] = -0.0389L + 2.836 | 0.52 |
| K | Hollow | [K] = -0.1488L + 10.224 | 0.47 |
| | Mound/polygon | [K] = -0.1159L + 8.119 | 0.33 |
| Al | Hollow | [Al] = 0.0555L - 3.3573 | 0.43 |
| | Mound/polygon | [Al] = 0.0577L - 3.4737 | 0.91 |
| Fe | Hollow | [Fe] = 0.1585L - 9.109 | 0.44 |
| | Mound/polygon | [Fe] = 0.1399L - 7.934 | 0.3 |
| Ti | Hollow | [Ti] = 0.462L - 27.841 | 0.52 |
| | Mound/polygon | [T] = 0.172L - 8.3533 | 0.19 |
| Mn | Hollow | [Mn] = 5.6454L - 351.11 | 0.41 |
| | Mound/polygon | [Mn] = 7.6632L - 481.39 | 0.44 |
| Co | Hollow | [Co] = 0.1618L - 9.9304 | 0.51 |
| | Mound/polygon | [Co] = 0.1658L - 10.218 | 0.5 |
| Ni | Hollow | [Ni] = 0.3096L - 18.437 | 0.43 |
| | Mound/polygon | [Ni] = 0.4012L - 24.19 | 0.55 |
| Cu | Hollow | [Cu] = 0.6695L - 39.754 | 0.54 |
| | Mound/polygon | [Cu] = 0.2503L - 10.948 | 0.63 |
| Zn | Hollow | [Zn] = -1.2677L + 90.571 | 0.56 |
| | Mound/polygon | [Zn] = -0.5584L + 44.424 | 0.78 |
| V | Hollow | [V] = 0.1308L - 7.9299 | 0.56 |
| | Mound/polygon | [V] = 0.2026L - 12.383 | 0.6 |
| Ga | Hollow | [Ga] = 0.0686L - 4.275 | 0.68 |
| | Mound/polygon | [Ga] = 0.0207L - 1.0605 | 0.03 |
| Rb | Hollow | [Rb] = -0.0229L + 1.939 | 0.11 |
| | Mound/polygon | [Rb] = -0.096L + 6.7962 | 0.48 |
| Cs | Hollow | [Cs] = -0.0036L + 0.2517 | 0.39 |
| | Mound/polygon | [Cs] = -0.0052L + 0.361 | 0.62 |
| Sr | Hollow | [Sr] = 0.7681L - 42.186 | 0.45 |
| | Mound/polygon | [Sr] = 1.2825L - 74.614 | 0.69 |
| Zr | Hollow | [Zr] = 0.0714L - 4.399 | 0.49 |
| | Mound/polygon | [Zr] = 0.0664L - 4.0544 | 0.57 |
| Mo | Hollow | [Mo] = -0.0116L + 0.8297 | 0.4 |
| | Mound/polygon | [Mo] = 0.0011L - 0.0092 | 0.01 |
| Sb | Hollow | [Sb] = -0.0068L+ 0.4819 | 0.53 |
| | Mound/polygon | [Sb] = -0.0069L + 0.489 | 0.54 |
| Cd | Hollow | [Cd] = -0.0919L + 6.1957 | 0.79 |
| | Mound/polygon | [Cd ]= -0.0402L + 2.8027 | 0.4 |
| La | Hollow | [La] = 0.0228L - 1.224 | 0.11 |
| | Mound/polygon | [La] = 0.0163L - 0.728 | 0.42 |
| Ce | Hollow | [Ce] = 0.0675L - 3.873 | 0.19 |

| | | Mound/polygon | [Ce] = 0.0387L - 1.8553 | 0.76 |
|---|---|---|---|---|
| Sm | Hollow | [Sm] = 0.0077L - 0.4591 | 0.34 |
| | Mound/polygon | [Sm] = 0.0084L - 0.4861 | 0.43 |
| Eu | Hollow | [Eu] = 0.0017L - 0.1001 | 0.56 |
| | Mound/polygon | [Eu] = 0.0015L - 0.0848 | 0.52 |
| Gd | Hollow | [Gd] = 0.0054L - 0.3021 | 0.24 |
| | Mound/polygon | [Gd] = 0.0094L - 0.5536 | 0.47 |
| Pr | Hollow | [Pr] = 0.008L - 0.4652 | 0.18 |
| | Mound/polygon | [Pr] = 0.0084L - 0.4788 | 0.46 |
| Dy | Hollow | [Dy] = -0.0003L + 0.0475 | 0.0004 |
| | Mound/polygon | [Dy] = -0.0057L + 0.41 | 0.4 |
| Yb | Hollow | [Yb] = 0.0032L - 0.189 | 0.49 |
| | Mound/polygon | [Yb] = 0.0038L - 0.2209 | 0.45 |
| Lu | Hollow | [Lu] = 0.0004L - 0.0202 | 0.39 |
| | Mound/polygon | [Lu] = 0.0006L - 0.0349 | 0.44 |
| W | Hollow | [W] = -0.0015L + 0.1214 | 0.049 |
| | Mound/polygon | [W] = -0.0038L + 0.2672 | 0.47 |
| Tl | Hollow | [Tl] = -0.0004L + 0.0327 | 0.11 |
| | Mound/polygon | [Tl] = -0.0015L + 0.1056 | 0.66 |
| Hf | Hollow | [Hf] = 0.0019L - 0.1135 | 0.47 |
| | Mound/polygon | [Hf] = 0.002L - 0.1187 | 0.68 |
| Pb | Hollow | [Pb] = -0.0438L + 3.5297 | 0.12 |
| | Mound/polygon | [Pb] = -0.1482L + 10.465 | 0.87 |
| Th | Hollow | [Th] = 0.0078L - 0.4603 | 0.34 |
| | Mound/polygon | [Th] = 0.0095L - 0.5465 | 0.92 |
| U | Hollow | [U] = 0.0021L - 0,1101 | 0.065 |
| | Mound/polygon | [U] = -0.0004L + 0,047 | 0.01 |













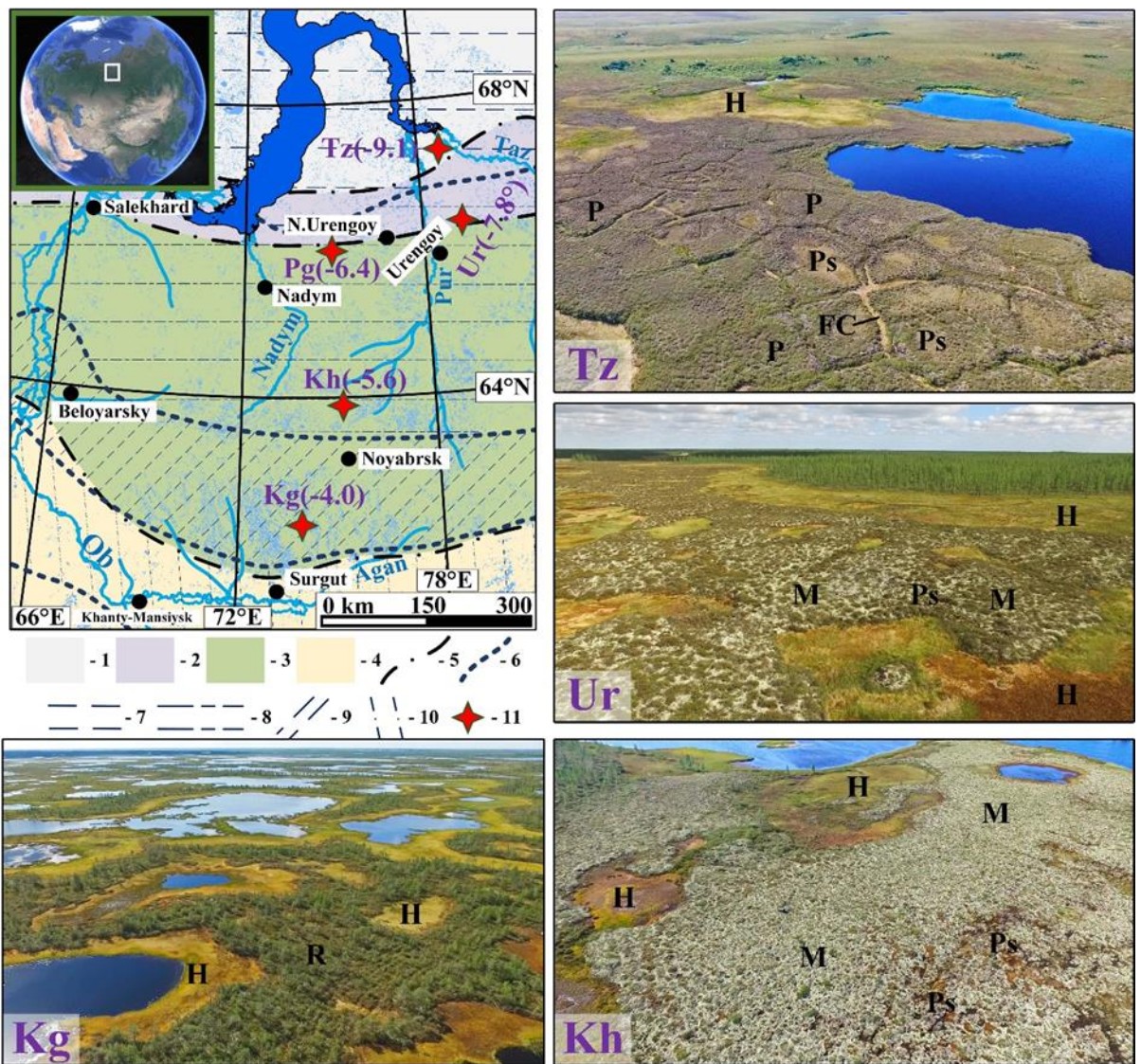



**Figure 1.** Map of the study site with permafrost boundaries (Brown et al., 2001; http://portal.inter-map.com (NSIDC)), with 5 main test sites: Kogalym (Kg), Khanymey (Kh), Pangody (Pg), Urengoy (Ur) and Tazovsky (Tz). The mean annual temperatures are given in parenthesis. The inserts represent aerial (drone-made) photos of main sites with the position of mound/polygon (M/P), hollow (H), frost crack (FC) and permafrost subsidence (Ps). On the Kogalym site, a hollow (H) – ridge (R) – lake complex is a dominating landscape type.

The numbers on the legend represent the following: 1, tundra; 2, forest-tundra; 3, northern taiga; 4, middle taiga; 5, borders between natural biomes; 6, borders between permafrost zones; 7, continuous permafrost; 8, discontinuous permafrost; 9, sporadic permafrost; 10, isolated permafrost; 11, key study sites with mean annual temperature in the parentheses.

1065
1066

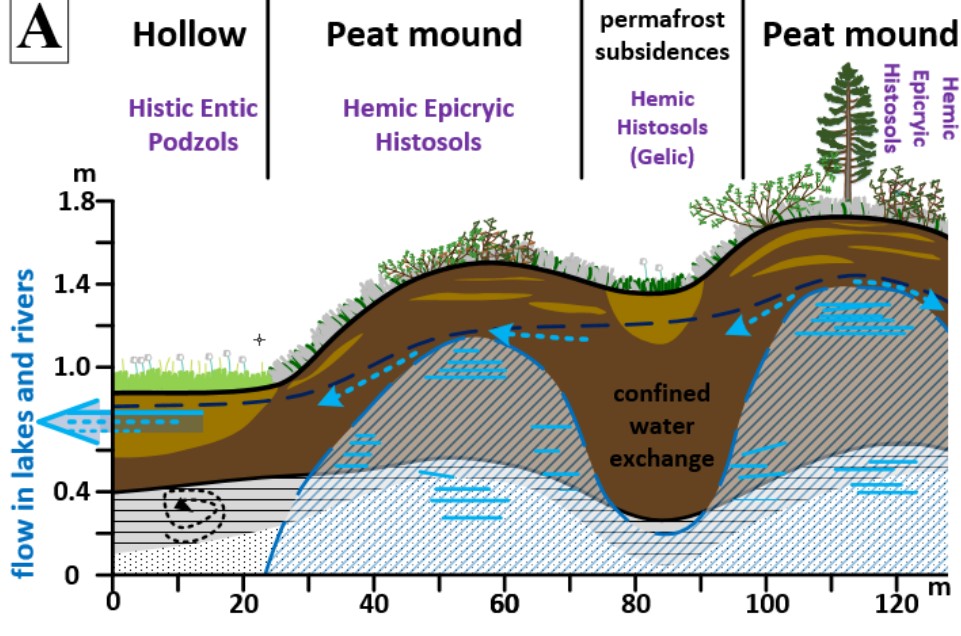

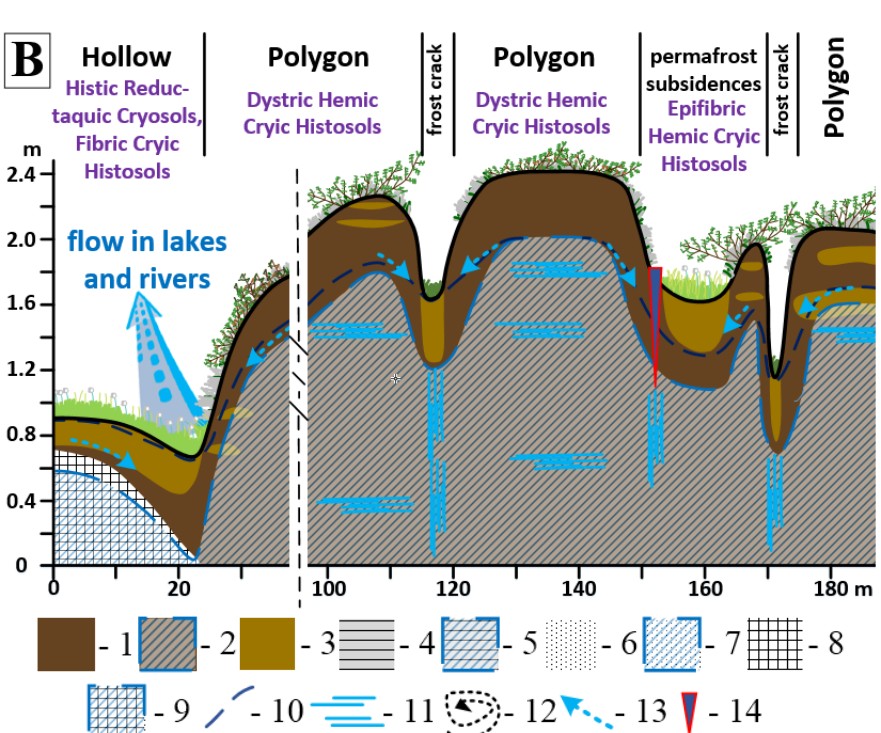

**Figure 2**. Soil transect of typical bog microlandscapes of flat mound palsa (A) and polygonal frozen bog (B). This vertical line in B indicates a discontinuity of hydrological flow-path. The numbers on the legend represent the following: 1, moss-lichen-sedge peat of medium degree of decomposition (Hemic); 2, permanently frozen peat; 3, moss-based peat of low degree of decomposition; 4, illuvial-Fe-humic (spodic) horizon; 5, permanently frozen spodic horizon; 6, sand and silt deposits; 7, frozen sand and silts; 8, heavy clay deposits; 9, frozen clays; 10, the level of suprapermafrost waters in August; 11, ice wedges; 12, cryoturbation features in soil; 13, the direction of soil water transport, typically along the permafrost boundary; 14, small crack on the polygonal bog.

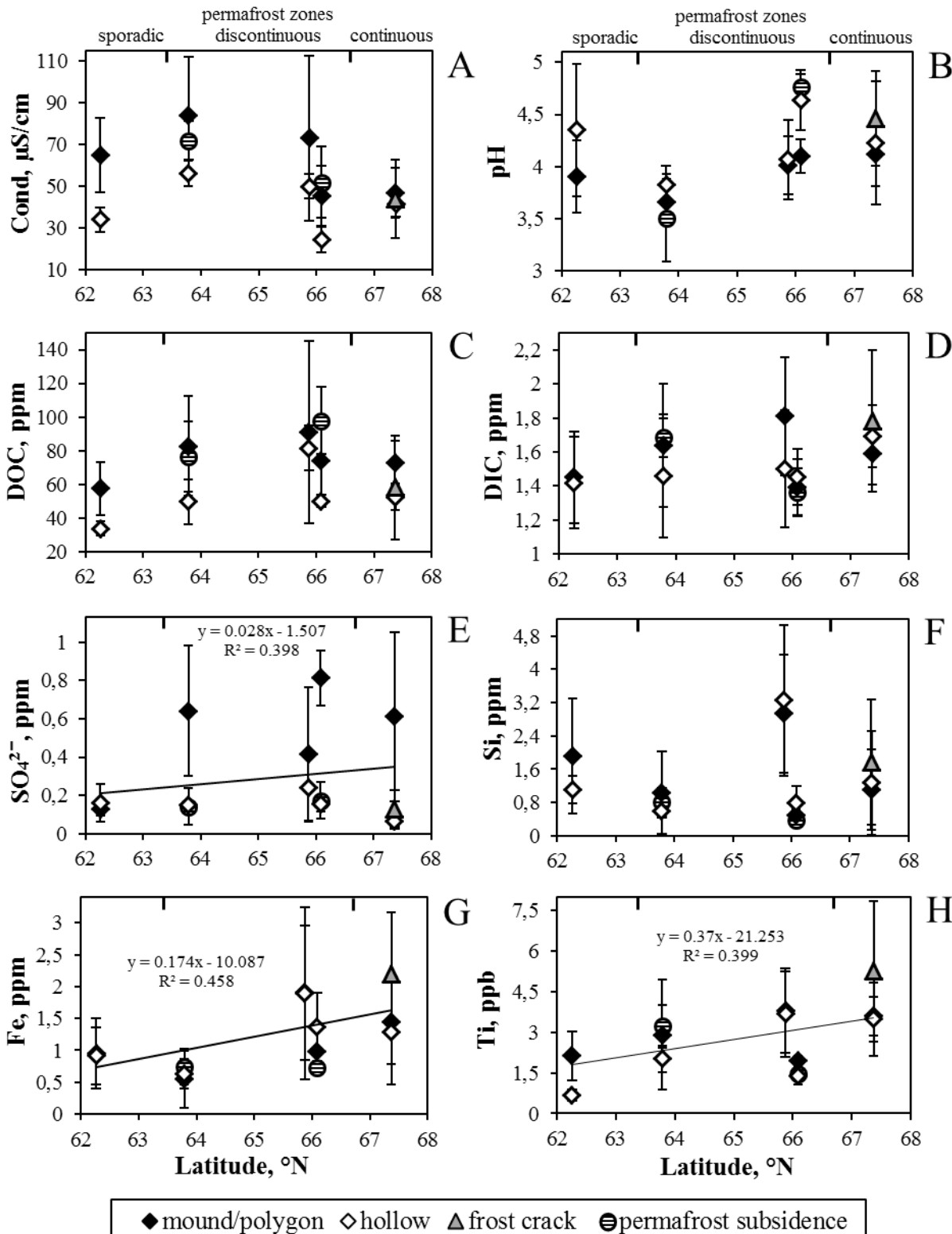

**Figure 3.** Mean values of Specific conductivity (A), pH (B), DOC (C), DIC (D), $SO_4^{2-}$ (E), Si (F), Fe (G) and Ti (H) concentration in peat porewaters of the WSL as a function of latitude for mound and polygons (solid diamonds), hollow (open diamonds), frost crack (grey triangles) and permafrost subsidence/depression (hatched circles). The solid line is a linear fit to all data with the regression equation given on each graph.

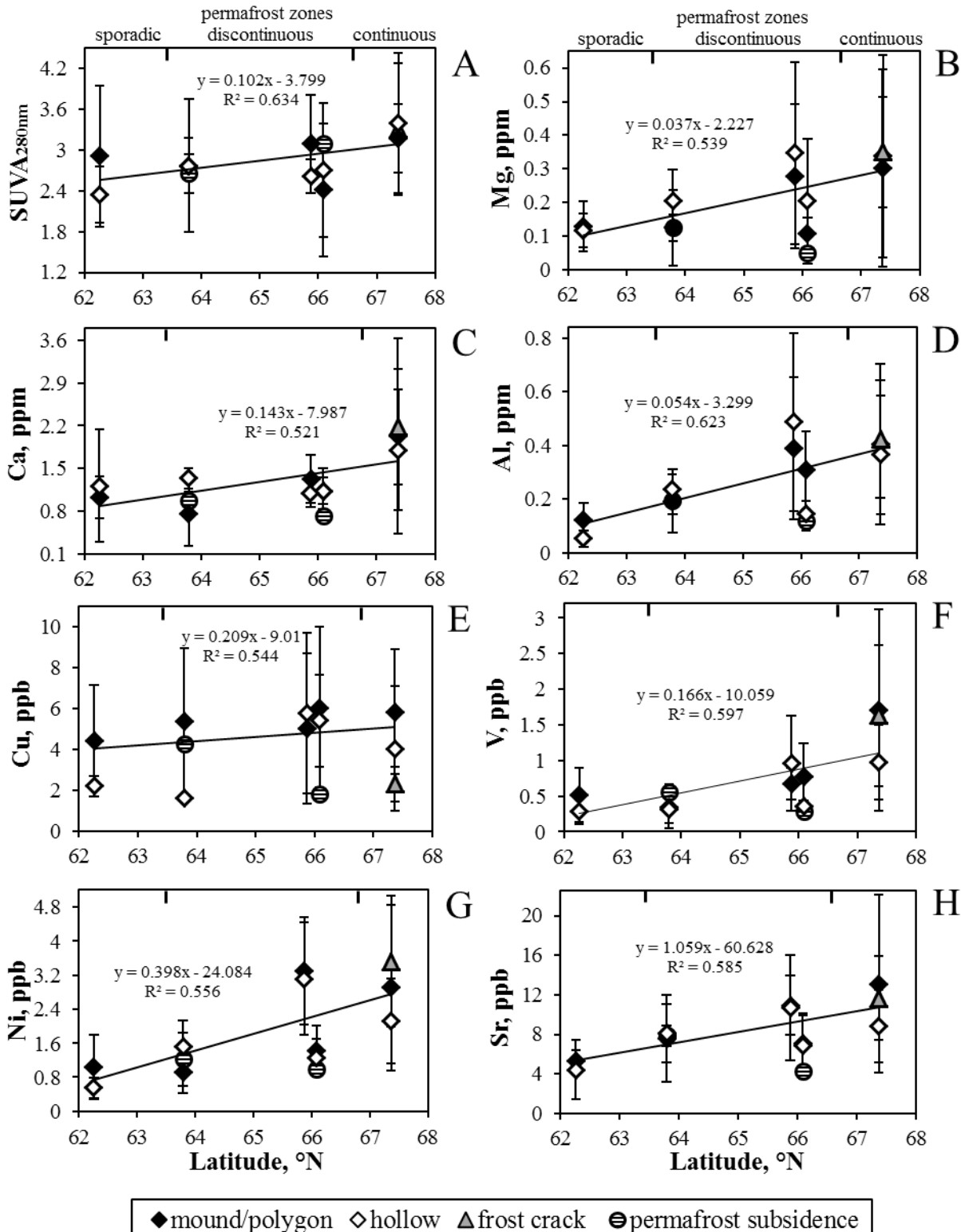

**Figure 4.** Mean values of $SUVA_{280}$ (A), Mg (B), Ca (C), Al (D), Cu (E), V (F), Ni (G), Sr (H) concentration in peat porewaters of the WSL as a function of latitude for mound and polygons (solid diamonds), hollow (open diamonds), frost crack (grey triangles) and permafrost subsidence/depression (hatched circles). The solid line is a linear fit to all data with the regression equation given on each graph.

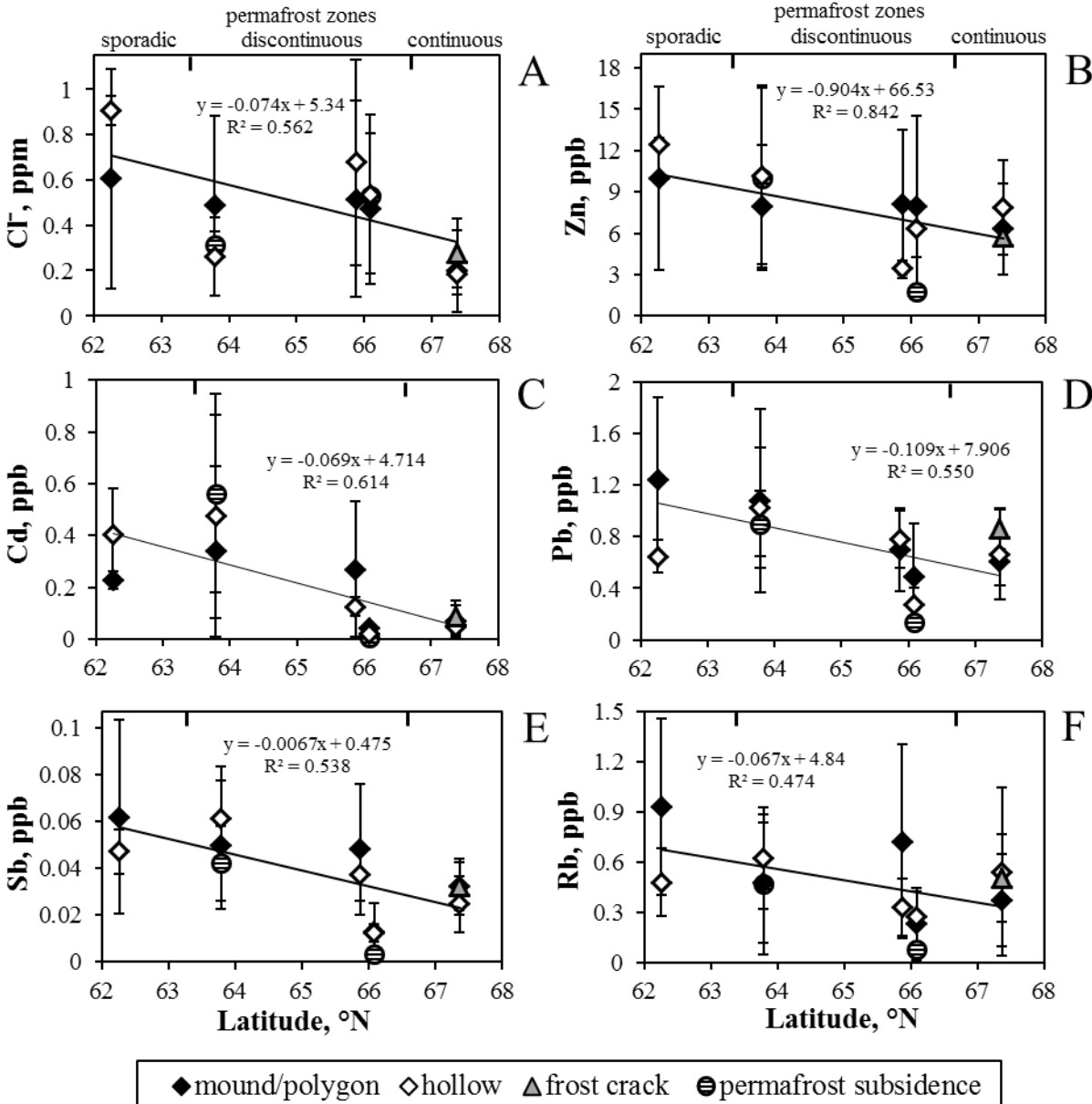

**Figure 5.** Mean concentrations of Cl (A), Zn (B), Cd (C), Pb (D), Sb (E), and Rb (F) in peat porewaters of the
WSL as a function of latitude for mound and polygons (solid diamonds), hollow (open diamonds), frost crack
(grey triangles) and permafrost subsidence/depression (hatched circles). The solid line is a linear fit to all data
with the regression equation given on each graph.