# Peer review of "Dissolved organic carbon, major and trace elements in peat pore water of sporadic, discontinuous and continuous permafrost zone of Western Siberia"

_Biogeosciences, 2017_

## Referee Comment (RC1) · Anonymous Referee #1 · 31 Mar 2017

Raudina and colleagues present an interesting and detailed study on pore-water sample data along an extensive latitudinal gradient in Western Siberia. The amount of data they have generated is substantial, but the way it is presented is reader-friendly by using a clear text structure and good statistical techniques. I was also glad to see an extensive and, to my knowledge, very complete and broad list of references throughout the text.

I certainly support publication of this manuscript, I think it adds valuable insights into the link between soils and aquatic systems, and potentially changing release pathways

upon future/ongoing permafrost thaw. I only have a few minor suggestions for revisions and a few thoughts to perhaps elaborate on:

Content: - Abstract: I would propose to either replace the final sentence, or add another one, something along the lines of line 510-512 (conclusions) to create an ending that is a bit more general - line 101: you here write that the precipitation gradient is from 400 to 460 mm but in Table 1 it ranges between 363 and 594mm? - is it possible to add one-two lines on the difference in origin for the two micro-landscapes you sketch out in Figure 2? - section 2.2: can you add some references for this method? - line 147-148: how many of the analyses did not show a good agreement? - lines 215-216: did you also consider comparing latitudinal gradients for mounds only, or for hollows only (instead of the average values per site independent of topgraphy)? - section 4.1: I am wondering: can the difference in DOC mounds vs. hollows also somehow be related to the (seasonal) timing of thaw? (Do the mounds thaw later than the hollows?) And hence the period of unfrozen exchange of constituents in the soil with porewater? Also, in line 379 you briefly mention that the chemical composition of peat between hollows and mounds may be different and could cause the differences in major and TE. Can this different chemical composition of peat not also play a role for the difference in DOC content between mounds and hollows? - line 257-259: this is an interesting statement and reference, but could you elaborate a bit more on how this relates to the above two sentences? - line 325-328: if DOC, Fe and Al are dominating colloidal carriers, why do none of the trace elements correlate to DOC? - lines 330-336: you present quite a lot of specific information/knowledge here, can you provide a bit better explanation so that more readers can follow? - lines 378-384: the difference in peat chemical composition is an important point, can you elaborate on this a bit more, also with respect to DOC patterns? - line 440-446: this is also an interesting paragraph, that I think you can expand a bit more. E.g., what can be the consequences of the correction for general (upscaling) calculations that are now made in literature? - line 467-468: I do not understand why the share of spring runoff from the mounds to rivers and lakes will decrease? And, perhaps related to this, have you considered any future

changes in precipitation patterns and/or general wetting/drying of the region? - line 474-476: here you present two scenarios that are presented as (i) OR (ii), but isn't it much more likely that both (i) AND (ii) will occur? - line 481: you write "proportion of mounds between 20 and 50%", is that a proportion of the total landscape? Or a proportion of the total elements? Please explain. - line 490-492: the fact that this study contradicts a dominating paradigm is something that can come forward a bit more, in my opinion, such as in the conclusions and/or in the abstract. - is there a reason why you measured SUVA280 and not the more commonly used SUVA254?

Tables and figures: - Table 1: write "latitude" instead of "GPS", and perhaps add the abbreviations for the regions (Tz, Ur, etc.) behind the site names - Figure 1: I think the panel with the actual map can be improved for increased readability, for example: enlarge picture, add either a vegetation map or biome map, or permafrost zonation map (instead of red lines) on the background (instead of the currently-used rather vague colours). Additionally, is it possible to add site maps with more detailed, high-res sampling locations of the different samples? - Figure 2: What is the vertical white line (with a dashed line in it) that crosses panel B through the left polygon? - Figure 3, 4, and 5: write "linear" instead of "liner". Also, it may be good to indicate the boundaries between the sporadic-discontinuous and discontinuous-continuous permafrost zones with vertical thin dashed lines?

Text edits/spelling: - Title: write "elements" instead of "element"? - line 55: "arctic" - line 156: "landscapes" - line 211: "pore waters" - I personally think ALT "rise" is not an ideal way of putting it, I would prefer to use ALT deepening or ALT thickening - Line 305-307: add "respectively" after this sentence - line 440: I suggest to write "our obtained results" - line 450: "in accordance" - line 464 and 466: "on the one hand" and "on the other hand" (not "from") - Olefeldt should be spelled throughout the manuscript with "dt"

In general, the language is quite good but I think the manuscript can benefit from a quick native-speaker check because particularly the use of articles ("the" and "a") is

often left out where it is required, and sometimes vice versa.

---

## Referee Comment (RC2) · Anonymous Referee #2 · 11 Apr 2017

The authors present an interesting dataset consisting of 80 porewater samples collected for 5 sites, spanning a latitudinal gradient in Western Siberia. Overall, I think the paper contains very useful results that may allow us to use a space for time approach in examining the potential changes in DOC/ DIC, trace-elemental and rare earth elemental compositions in pore waters following deepening AL.

The overall structure is sensible, yet the text is slightly unclear in places likely simply due to language barriers.

My main concern with the manuscript is how the statistics have been conducted and

with the potential for improvements to be made in the analysis and then potentially the interpretation. For example, how were each of the variables normalised before the PCA was conducted and can you demonstrate that the PCA results explain a significant proportion of the variance in your dataset? I think you show that the 2 extracted PCA axes explain only 29% of the total variance in the dataset? Would you not be better served trying to improve this, or using a prior step to remove variables that do not show significant differences between sites (e.g. using a Kruskal-Wallis test). Including only the significant data may improve the PCA and allow for wider patterns to be identified. You may be advised to apply a Kaiser-Meyer-Olkin (KMO) Measure of Sampling Adequacy for the overall data set to detect if you have sufficient sampling adequacy to include all of these parameters together. Another approach, maybe to break the dataset down and conduct separate PCA. Also, how did you define the number of eigenfactors you chose for the PCA?

Furthermore, the study often uses multiple Mann-Whitney U tests between pairs of variables, even when a more appropriate approach maybe to use a test that allows more than 2 parameters to be compared (e.g. one-way ANOVA on ranks/ Kruskal-Wallis).

Overall, I feel the study was well conducted and worthy of publication, yet at the moment I feel the statistical approaches need to be re-addressed (or explained more clearly if these approaches are not valid?).

Additional comments: 15 - "is one of the major consequences" currently unclear. 16 - deepening not "rise" 21 - "expected decrease" why would you expect the intensity of DOC and TE mobilisation to decrease? I think you need to provide a rationale for this hypothesis earlier on. 27 - Need to define REEs on first use. What does this stand for (rare earth elements?) and what are you including in this group? actual values ? 33-36 "will not exceed 20%" actual values - do you mean they will not change from current values? 55- misspelt arctic 65-67 - please check these references with what you are referring too. For example, I do not think Mann et al 2015 examines lakes,

or Vonk et al. 2015b soil leachates? Also, if you are discussing soil porewaters, you should likely include: "Optical properties and bioavailability of dissolved organic matter along a flow-path continuum from soil pore waters to the Kolyma River mainstem, East Siberia Frey et al. Biogeosciences"

General - should be consistent with use of either trace element or TE throughout.

82 - elements 86 - "feeding of" is awkward, maybe use "source to" 87 - reference needed for this statement. 152 - its not clear to me where you used the ANOVA in the results - maybe I missed it? 158-162 Here is where I think we need far more info on the approach used in the PCA. 161 - I don't think it really acts upon (as this suggests its constrained in some way) rather it 'explains' a greater variance in. 162 - Need more information here on how you used the PCA to test the influence of lat and ALT on DOC and TE. Were you planning to relating the PC loadings or running constrained PCAs?

168 - Unclear to me how this PCA was run. Did you normalise and standardise each of the measurements? This can have a dramatic effect upon PCA loadings and the potential weighting effects of each measurement. 169 - I don't think it really acts upon (as this suggests its constrained in some way) rather it 'explains' a greater variance in. 173-176 - Does this mean all of the these were significant to «0.05 and have greater R value of 0.6?> In associated supplemental - can you example what the coloured arrows on this plot refer to? and what W stands for? 206 - this is unclear to me. So you did separate tests for each possible site? I think you should run one capable of testing for overall differences first and then examining significant differences. 216 - So why show linear regressions? I would only add these if there is are significant differences between at least the two endmember sites. Adding them to all of the graphs just makes its harder to see the values and error bars. Fig S3 figure text needs an explanation of the circled areas of A. Also, are both not containing the same explanatory variables and neither showing site loadings? 228 - does this mean that R-values varied between these? This way of showing the range is unusual to me. 243 - were these also SUVA 280? As most studies use SUVA at 254nm. 249-256 - I think a range of 2 to 3.5

for SUVA is actually very large. For SUVA 254 for example, we may only expect a natural variation of between 1.5 to 5.5 in pore to coastal waters (in Eastern Siberian freshwaters). A change from 2 to 3.5 demonstrates a significant shift in the composition of the DOM and will have a pronounced effect upon the biogeochemical processing of DOM upon export.

Leachates of permafrost and active layer peats will also demonstrate clearly that SUVA254 at least id much lower in permafrost material across at least most parts of Siberia and Alaska that have been studied. Would the values you have collected not simply be indicative of collecting waters predominately sources from active layer soils and limited permafrost thaw influence?

Could the higher SUVA further North not instead be suggestive of lower rates of C processing within soil environments?

399 - "incoming into" may be instead "prior to export to"

---

## Author Comment (AC1) · 30 Apr 2017

*The reviewer stated that "Overall, I feel the study was well conducted and worthy of publication, yet at the moment I feel the statistical approaches need to be re-addressed (or explained more clearly if these approaches are not valid?"*
We appreciate the positive evaluation of our work and we greatly revised statistical treatment following the recommendations of reviewer as described below.

*The main concern of this reviewer with the manuscript is how the statistics have been conducted and with the potential for improvements to be made in the analysis and then potentially the interpretation.* A lot of new statistical treatments were performed (PCA, Kruskal-Wallis). Results are presented in this reply. The essential of the interpretation has not been modified.

*For example, how were each of the variables normalized before the PCA was conducted and can you demonstrate that the PCA results explain a significant proportion of the variance in your dataset?* The identification of factors was performed using the method of Raw Data and the extraction method was principal component. All the variables were normalized as necessary in standard package of Statistica-7 given that the units of various components are different. We do not expect that the PCA is capable explaining high proportion of variance in all major and trace element concentration. Given the high number of variables and the diversity of environmental conditions responsible for soil solution composition formation, 20 and 9 % of total variation is not a bad result. Note that the PCA treatment of the river water in western Siberia also allowed explanation of "only" 20 and 10% of the variance in a much larger dataset (Pokrovsky et al., 2016a).

*I think you show that the 2 extracted PCA axes explain only 29% of the total variance in the dataset? Would you not be better served trying to improve this, or using a prior step to remove variables that do not show significant differences between sites (e.g. using a Kruskal-Wallis test). Including only the significant data may improve the PCA and allow for wider patterns to be identified.* This is very pertinent comment. We did attempt to remove part of the components (variables) in order to improve the PCA. This exercise was not successful because the number of individual measurements was not sufficiently high.

*You may be advised to apply a Kaiser-Meyer-Olkin (KMO) Measure of Sampling Adequacy for the overall data set to detect if you have sufficient sampling adequacy to include all of these parameters together.* Good point. Unfortunately, in this study we used standard STATISTICA-7 package which, unlike SPSS, does not allow realization of Kaiser-Meyer-Olkin (KMO) criterion. Nevertheless, following this useful advice, we computed the KMO criterion using Excel. As we expected, the KMO value was equal to 0.533 which suggests rather low adequacy: the analysis does not make sense at KMO $< 0.5$.

*Another approach, maybe to break the dataset down and conduct separate PCA.* We tried this: removal a part of the data series and conducting separate PCA for major elements, TE, various forms of micro-relief and various geographical sites did not yield any better description because of insufficient size of the dataset.

*Also, how did you define the number of eigenfactors you chose for the PCA?* The number of maximal number of eigenfactors was based on the criterion of PCA values decrease. We used a scree test for determining the number of factors to retain in a factor analysis or principal components analysis. The scree test involves plotting the eigenvalues in descending order of their magnitude against their factor numbers and determining where they level off.
The PCA values demonstrated significant decrease of the values between F2 and F3 (**Fig. 1R** of this Reply) suggesting therefore that at least two factors are interpretable.

[Figure]

**Fig. 1R**. The PCA value as a function of the number of eigenvalues.

This produced the following table of eigenvalues:

| Eigenvalues Extraction: Principal components | | | |
|---|---|---|---|
| | **Eigenvalue** | **% Total** | **Cumulative** | **Cumulative** |
| **1** | 21,84690 | 39,01 | 21,8469 | 39,012 |
| **2** | 7,42489 | 13,26 | 29,2718 | 52,271 |

*Furthermore, the study often uses multiple Mann-Whitney U tests between pairs of variables, even when a more appropriate approach maybe to use a test that allows more than 2 parameters to be compared (e.g. one-way ANOVA on ranks/ KruskalWallis).*

This is really important advice and we directly followed this recommendation. Major and TE concentrations in soil porewaters of (1) five main sampling sites and (2) four main micro-relief landscapes (polygon, permafrost/subsidence, frost crack and hollow) were additionally processed using nonparametric H-criterion Kruskal–Wallis test. This test is suitable for evaluation of difference of each component among several samplings simultaneously. It is considered statistically significant at $p < 0.05$. Results of two tests are listed in **Tables R1** and **R2** below. For all components and micro-landscapes, we observed full consistency between Kruskal–Wallis and Mann–Whitney U tests.

**Table R1.** Statistical differences in elements concentration between different forms of microrelief for each key sites. p-values are determined by Mann–Whitney U test (1st line) and Kruskal–Wallis test (2nd line) for each component. Note that Kogalym and Pangody sites present only Mann–Whitney U test because they have only two forms of microrelief (mound and hollow).

| Component | Kogalym | Khanymey | | | Pangody | Urengoy | | | Tazovskiy | | |
|---|---|---|---|---|---|---|---|---|---|---|---|
| | mound – hollow | mound – hollow | mound – permafrost subsidence | hollow – permafrost subsidence | mound – hollow | mound – hollow | mound – permafrost subsidence | hollow – permafrost subsidence | polygon – hollow | polygon – frost crack | hollow – frost crack |
| Cond | **0.035** | **0.032** | 0.441 | 0.083 | 0.107 | **0.034** | 0.564 | **0.029** | 0.387 | 0.544 | 0.302 |
| | | H = 6.20; **p = 0.045** | | | | H = 6.11; **p = 0.047** | | | H = 0.879; p = 0.644 | | |
| pH | 0.519 | 0.114 | 0.685 | 0.312 | 0.693 | **0.050** | **0.048** | 0.355 | 0.592 | 0.130 | 0.302 |
| | | H = 2.57; p = 0.277 | | | | H = 6.3; **p = 0.043** | | | H = 2.257; p = 0.324 | | |
| Cl⁻ | 0.086 | 0.292 | 0.465 | 0.564 | 0.294 | 0.724 | 0.564 | 0.729 | 0.435 | 0.182 | 0.121 |
| | | H = 1.492; p = 0.474 | | | | H = 0.278; p = 0.87 | | | H = 2.038; p = 0.361 | | |
| $SO_4^{2-}$ | 0.238 | **0.028** | **0.015** | 0.072 | **0.038** | **0.034** | **0.028** | 0.749 | **0.016** | **0.018** | **0.025** |
| | | H = 9.21; **p = 0.01** | | | | H = 6.975; **p = 0.031** | | | H = 15.568; **p = 0.0004** | | |
| DOC | **0.043** | **0.023** | 0.283 | **0.049** | 0.082 | **0.037** | **0.048** | **0.046** | **0.027** | **0.033** | 0.535 |
| | | H = 6.291; **p = 0.043** | | | | H = 6.475; **p = 0.039** | | | H = 8.206; **p = 0.017** | | |
| DIC | 0.643 | 0.194 | 0.626 | **0.017** | **0.031** | 0.485 | 0.718 | 0.157 | 0.093 | 0.140 | 0.180 |
| | | H = 6.103; **p = 0.047** | | | | H = 0.078; p = 0.962 | | | H = 1.492; p = 0.474 | | |
| Ca | 0.479 | **0.043** | 0.256 | **0.017** | 0.304 | 0.067 | **0.042** | **0.043** | 0.195 | 0.124 | 0.540 |
| | | H = 6.66; **p = 0.036** | | | | H = 6.475; **p = 0.039** | | | H = 0.37; p = 0.831 | | |
| Mg | 0.542 | **0.029** | 0.639 | **0.044** | 0.641 | **0.048** | **0.045** | **0.041** | 0.730 | 0.390 | 0.530 |
| | | H = 6.315; **p = 0.042** | | | | H = 6.291; **p = 0.043** | | | H = 0.877; p = 0.645 | | |
| K | 0.157 | 0.271 | 0.631 | 0.164 | 0.132 | 0.097 | **0.036** | **0.049** | 0.320 | 0.210 | 0.180 |
| | | H = 0.083; p = 0.959 | | | | H = 5.468; **p = 0.047** | | | H = 3.531; p = 0.171 | | |
| Al | **0.046** | **0.047** | 0.517 | **0.043** | 0.082 | **0.047** | **0.048** | 0.157 | **0.049** | 0.058 | 0.540 |
| | | H = 6.9; **p = 0.032** | | | | H = 6.3; **p = 0.043** | | | H = 5.468; **p = 0.047** | | |
| Fe | **0.048** | **0.046** | **0.043** | 0.234 | 0.634 | **0.039** | **0.048** | **0.031** | **0.048** | **0.029** | **0.042** |
| | | H = 6.568; **p = 0.038** | | | | H = 6.737; **p = 0.034** | | | H = 7.606; **p = 0.022** | | |
| Si | **0.039** | 0.283 | 0.221 | 0.308 | **0.045** | **0.04** | 0.363 | **0.043** | 0.554 | **0.032** | **0.048** |
| | | H = 1.636; **p = 0.44** | | | | H = 6.275; **p = 0.043** | | | H = 6.522; **p = 0.038** | | |
| Li | **0.029** | 0.192 | 0.746 | 0.564 | 0.638 | **0.047** | 0.818 | 0.050 | **0.045** | 0.054 | 0.053 |
| | | H = 1.15; p = 0.563 | | | | H = 6.112; **p = 0.047** | | | H = 6.44; **p = 0.038** | | |
| B | **0.038** | **0.039** | **0.029** | 0.386 | 0.221 | – | – | – | **0.023** | 0.098 | 0.074 |
| | | H = 7.01; **p = 0.03** | | | | – | | | H = 10.258; **p = 0.006** | | |
| Na | 0.397 | 0.194 | 0.265 | 0.248 | 0.063 | 0.289 | 0.083 | 0.064 | 0.102 | 0.506 | 0.202 |
| | | H = 2.01; p = 0.367 | | | | H = 2.8; p = 0.247 | | | H = 1.509; p = 0.47 | | |
| Ti | **0.031** | 0.441 | 0.156 | 0.083 | 0.453 | 0.157 | 0.248 | 0.355 | 0.654 | 0.066 | 0.091 |
| | | H = 2.56; p = 0.278 | | | | H = 4.54; p = 0.103 | | | H = 0.959; p = 0.619 | | |

| Element | | | | | | | | | | | |
|---|---|---|---|---|---|---|---|---|---|---|---|
| V | 0.086 | 0.570 | 0.330 | 0.083 | 0.267 | **0.037** | **0.026** | 0.443 | 0.134 | 0.467 | 0.302 |
| | | H = 1.977; p = 0.372 | | | | H = 7; **p = 0.03** | | | H = 1.324; p = 0.516 | | |
| Cr | 0.053 | 0.521 | 0.465 | 0.172 | 0.221 | 0.157 | 0.564 | 0.064 | 0.676 | 0.544 | 0.339 |
| | | H = 0.438; p = 0.803 | | | | H = 2.5; p = 0.286 | | | H = 0.641; p = 0.726 | | |
| Mn | 0.091 | **0.046** | **0.044** | 0.064 | **0.031** | **0.037** | **0.048** | 0.095 | 0.108 | 0.476 | 0.239 |
| | | H = 6.78; **p = 0.034** | | | | H = 6.051; **p = 0.048** | | | H = 2.177; p = 0.337 | | |
| Co | 0.283 | 0.144 | **0.043** | 0.386 | 0.307 | 0.480 | 0.564 | 0.165 | 0.532 | 0.090 | 0.121 |
| | | H = 6.283; **p = 0.043** | | | | H = 1.94; p = 0.378 | | | H = 3.188; p = 0.203 | | |
| Ga | 0.053 | 0.05 | 0.775 | **0.021** | **0.041** | 0.289 | 0.083 | 0.355 | 0.053 | 0.052 | **0.046** |
| | | H = 6.23; **p = 0.044** | | | | H = 3.3; p = 0.192 | | | H = 6.05; **p = 0.048** | | |
| As | 0.190 | **0.022** | 0.023 | 0.148 | 0.074 | 0.624 | **0.046** | 0.101 | 0.312 | 0.115 | 0.058 |
| | | H = 6.131; **p = 0.047** | | | | H = 6.05; **p = 0.048** | | | H = 3.548; p = 0.169 | | |
| Rb | 0.043 | 0.072 | 0.808 | 0.564 | **0.041** | 0.480 | **0.038** | **0.046** | **0.049** | 0.052 | 0.614 |
| | | H = 0.823; p = 0.663 | | | | H = 6.14; **p = 0.046** | | | H = 5.968; **p = 0.050** | | |
| Zr | **0.032** | 0.570 | 0.256 | 0.149 | 0.053 | 0.706 | 0.148 | 0.063 | 0.095 | 0.467 | 0.108 |
| | | H = 2.044; p = 0.359 | | | | H = 0.811; p = 0.666 | | | H = 1.23; p = 0.54 | | |
| Nb | **0.048** | 0.168 | 0.746 | 0.564 | 0.414 | 0.527 | 0.564 | 0.455 | 0.284 | 0.782 | 0.210 |
| | | H = 1.817; p = 0.403 | | | | H = 1; p = 0.607 | | | H = 0.964; p = 0.618 | | |
| Mo | **0.042** | 0.317 | 0.144 | 0.441 | 0.579 | 0.724 | 0.585 | 0.643 | 0.272 | **0.037** | **0.020** |
| | | H = 2.38; p = 0.311 | | | | H = 0.1 p = 0.95 | | | H = 6.48; **p = 0.039** | | |
| Cd | **0.032** | 0.105 | **0.037** | 0.342 | **0.044** | **0.029** | **0.023** | 0.052 | **0.044** | 0.132 | 0.233 |
| | | H = 6.568; **p = 0.038** | | | | H = 7; **p = 0.03** | | | H = 6.05; **p = 0.048** | | |
| Ni | 0.147 | **0.044** | 0.162 | 0.381 | 0.732 | 0.057 | 0.560 | 0.408 | 0.446 | 0.467 | 0.089 |
| | | H = 6.045; **p = 0.0487** | | | | H = 1.34; p = 0.511 | | | H = 2.128; p = 0.345 | | |
| Cu | **0.035** | **0.028** | 0.268 | **0.018** | 0.641 | 0.485 | **0.028** | **0.027** | 0.128 | **0.029** | **0.036** |
| | | H = 7.408; **p = 0.025** | | | | H = 6.437; **p = 0.038** | | | H = 6.737; **p = 0.034** | | |
| Zn | 0.479 | 0.372 | 0.372 | 0.734 | **0.021** | 0.720 | **0.038** | **0.037** | 0.270 | 0.740 | 0.250 |
| | | H = 1.373; p = 0.503 | | | | H = 7; **p = 0.03** | | | H = 1.648; p = 0.439 | | |
| Sr | 0.358 | 0.516 | 0.424 | 0.712 | 0.571 | 0.512 | 0.183 | 0.094 | **0.047** | 0.762 | 0.345 |
| | | H = 1.54; p = 0.463 | | | | H = 2.24; p = 0.326 | | | H = 6.05; **p = 0.049** | | |
| Sb | 0.519 | 0.224 | 0.746 | 0.248 | 0.480 | 0.480 | **0.048** | **0.040** | 0.176 | 0.808 | 0.302 |
| | | H = 1.788; p = 0.409 | | | | H = 6.141; **p = 0.046** | | | H = 1.991; p = 0.369 | | |
| Cs | 0.667 | 0.681 | 0.685 | 0.773 | 0.307 | 0.289 | 0.564 | 0.255 | 0.052 | **0.018** | **0.012** |
| | | H = 0.21; p = 0.9 | | | | H = 1.61; p = 0.447 | | | H = 9.9; **p = 0.007** | | |
| Ba | 0.083 | 0.675 | 0.426 | 0.703 | 0.105 | 0.089 | **0.048** | 0.155 | 0.781 | 0.225 | 0.197 |
| | | H = 0.053; p = 0.974 | | | | H = 6.112; **p = 0.047** | | | H = 1.684; p = 0.431 | | |
| La | 0.133 | **0.046** | **0.041** | 0.386 | 0.130 | **0.045** | **0.041** | 0.307 | 0.091 | 0.544 | **0.039** |
| | | H = 6.278; **p = 0.043** | | | | H = 6.141; **p = 0.046** | | | H = 6.275; **p = 0.043** | | |
| Ce | **0.048** | 0.685 | 0.685 | 0.386 | 0.414 | **0.046** | **0.038** | 0.343 | 0.176 | 0.587 | 0.097 |
| | | H = 0.391; p = 0.823 | | | | H = 6.475; **p = 0.039** | | | H = 2.31; p = 0.316 | | |
| Pr | **0.043** | 0.165 | 0.246 | 0.106 | **0.044** | 0.157 | 0.248 | 0.543 | 0.108 | 0.674 | 0.071 |

| | | | | | | | | | | | |
|---|---|---|---|---|---|---|---|---|---|---|---|
| | H = 0.716; p = 0.699 | | | | | H = 1.84; p = 0.398 | | | H = 3.681159 p = 0.1587 | | |
| Nd | **0.032** | 0.208 | 0.226 | 0.386 | 0.540 | **0.034** | 0.053 | 0.073 | 0.094 | 0.875 | 0.121 |
| | H = 0.246; p = 0.884 | | | | | H = 5.97; **p = 0.051** | | | H = 3.29; p = 0.193 | | |
| Sm | **0.032** | 0.417 | 0.146 | 0.248 | 0.535 | 0.289 | 0.248 | 0.556 | 0.105 | 0.853 | 0.097 |
| | H = 0.94; p = 0.625 | | | | | H = 1.84; p = 0.398 | | | H = 3.29; p = 0.193 | | |
| Eu | **0.043** | 0.064 | 0.087 | 0.328 | 0.838 | 0.289 | 0.232 | 0.643 | **0.043** | 0.396 | **0.047** |
| | H = 0.744; p = 0.689 | | | | | H = 1.84; p = 0.398 | | | H = 6.112; **p = 0.047** | | |
| Gd | 0.133 | 0.685 | 0.113 | 0.248 | 0.540 | 0.089 | **0.038** | 0.243 | **0.046** | 0.822 | 0.197 |
| | H = 0.378; p = 0.828 | | | | | H = 6.475; **p = 0.039** | | | H = 6.141; **p = 0.046** | | |
| Tb | 0.086 | **0.042** | **0.015** | 0.128 | 0.414 | 0.089 | **0.048** | 0.343 | **0.043** | 0.716 | 0.107 |
| | H = 8.229; **p = 0.016** | | | | | H = 6.051; **p = 0.049** | | | H = 5.967; **p = 0.05** | | |
| Dy | **0.048** | 0.385 | 0.187 | 0.248 | 0.221 | 0.128 | **0.042** | 0.720 | **0.046** | 0.628 | 0.057 |
| | H = 0.378; p = 0.828 | | | | | H = 6.05; **p = 0.049** | | | H = 6.395; **p = 0.041** | | |
| Ho | 0.086 | 0.771 | 0.372 | 0.473 | 0.540 | 0.359 | **0.048** | 0.643 | 0.053 | 0.828 | 0.067 |
| | H = 0.011; p = 0.99 | | | | | H = 1.42; **p = 0.049** | | | H = 3.758; p = 0.153 | | |
| Er | **0.043** | 0.775 | 0.871 | 0.128 | 0.783 | 0.480 | 0.068 | 0.533 | 0.108 | 0.628 | 0.065 |
| | H = 0.244; p = 0.885 | | | | | H = 1.34; p = 0.51 | | | H = 3.165; p = 0.21 | | |
| Tm | 0.086 | 0.062 | 0.187 | 0.248 | 0.740 | 0.480 | 0.098 | 0.546 | 0.148 | 0.889 | 0.121 |
| | H = 0.172; p = 0.92 | | | | | H = 1.34; p = 0.511 | | | H = 3.29; p = 0.193 | | |
| Yb | 0.053 | 0.626 | 0.708 | 0.248 | 0.767 | 0.512 | **0.037** | 0.146 | 0.51 | 0.808 | 0.302 |
| | H = 0.599; p = 0.74 | | | | | H = 6.522; **p = 0.038** | | | H = 3.548; p = 0.169 | | |
| Lu | 0.086 | 0.111 | 0.181 | 0.386 | 0.844 | 0.724 | 0.248 | 0.494 | 0.128 | 0.840 | 0.097 |
| | H = 0.083; p = 0.959 | | | | | H = 1; p = 0.607 | | | H = 2.794; p = 0.247 | | |
| Hf | **0.032** | **0.029** | 0.292 | 0.563 | 0.414 | 0.562 | 0.091 | 0.147 | 0.399 | 0.467 | 0.121 |
| | H = 7.335; **p = 0.026** | | | | | H = 1; p = 0.607 | | | H = 1.592; p = 0.45 | | |
| W | 0.519 | 0.716 | 0.372 | 0.423 | 0.556 | **0.048** | **0.021** | **0.024** | 0.612 | 0.220 | 0.079 |
| | H = 0.637; p = 0.727 | | | | | H = 7.336; **p = 0.026** | | | H = 0.732; p = 0.694 | | |
| Tl | 0.378 | 0.042 | 0.746 | 0.564 | 0.683 | 0.157 | 0.265 | 0.064 | 0.838 | **0.025** | **0.039** |
| | H = 0.744; p = 0.689 | | | | | H = 4.2; p = 0.123 | | | H = 6.467; **p = 0.039** | | |
| Pb | **0.048** | 0.082 | 0.081 | 0.264 | 0.093 | **0.048** | **0.038** | **0.046** | 0.615 | **0.039** | **0.035** |
| | H = 0.117; p = 0.943 | | | | | H = 6.144; **p = 0.046** | | | H = 6.467; **p = 0.039** | | |
| Th | 0.053 | 0.072 | 0.087 | 0.248 | 0.827 | 0.096 | 0.056 | 0.130 | 0.076 | 0.399 | 0.121 |
| | H = 1.103; p = 0.576 | | | | | H = 1.11; p = 0.574 | | | H = 3.766; p = 0.152 | | |
| U | 0.667 | 0.094 | 0.087 | 0.248 | **0.044** | **0.046** | 0.064 | 0.063 | 0.351 | 0.467 | 0.302 |
| | H = 0.186; p = 0.91 | | | | | H = 6.395; **p = 0.041** | | | H = 2.379; p = 0.304 | | |

**Table R2**. Statistical differences of elements concentration between sites in peat soil solutions within different micro-landscapes. p-values are determined first by Kruskal–Wallis H-test and then by Mann–Whitney U test for each component.

| Study site | Chemical elements, p-value is determined by Wilcoxon-Mann Whitney test | | | | | | | | | | | | | | | | | | | | | | | | | |
|---|---|---|---|---|---|---|---|---|---|---|---|---|---|---|---|---|---|---|---|---|---|---|---|---|---|---|
| | DOC | DIC | Ca | Mg | K | Na | Si | Al | Fe | Zn | Li | Cu | Ni | Sr | Mn | Rb | As | Co | Cd | Pb | Ba | La | Nd | Yb | Th | U |
| **Mound/polygon** | | | | | | | | | | | | | | | | | | | | | | | | | | |
| Results of all site comparison on the mound/polygon for each component (Kruskal–Wallis H-test) | | | | | | | | | | | | | | | | | | | | | | | | | | |
| Kogalym Khanymey Pangody Urengoy Tazovskiy | H = 10.101 p = 0.039 | H = 9.851 p = 0.043 | H = 11.91 p = 0.018 | H =10.07 p = 0.039 | H = 23.109 p = 0.0001 | H = 9.611 p = 0.048 | H =11.59 p = 0.021 | H =12.56 p =0.014 | H =18.91 p =0.001 | H = 9.768 p = 0.046 | H = 18.562 p = 0.001 | H = 9.564 p = .048 | H = 24.36 p = 0.0001 | H = 9.795 p = 0.044 | H = 33.17 p = 0.0001 | H = 9.641 p = 0.047 | H =10.94 p = 0.027 | H = 31.91 p = 0.0001 | H = 9.909 p = 0.042 | H = 9.666 p = 0.047 | H = 9.933 p = 0.0042 | H =1.395 p = 0.845 | H =10.49 p = 0.033 | H =10.11 p = 0.037 | H = 9.658 p = 0.0466 | H = 10.269 p = 0.036 |
| Pairwise comparison of key sites for each component (Mann–Whitney U test) | | | | | | | | | | | | | | | | | | | | | | | | | | |
| Kogalym | | | | | | | | | | | | | | | | | | | | | | | | | | |
| Khanymey | **0.035** | 0.315 | **0.033** | 0.516 | **0.009** | **0.044** | **0.042** | 0.277 | **0.041** | 0.461 | 0.074 | 0.646 | **0.048** | 0.570 | **0.029** | 0.052 | **0.033** | 0.746 | 0.256 | 0.570 | 0.224 | 0.626 | 0.212 | 0.516 | 0.516 | 0.57 |
| Pangody | **0.028** | 0.081 | **0.012** | 0.231 | 0.395 | 0.534 | 0.256 | **0.018** | 0.234 | 0.496 | 0.092 | 0.645 | **0.017** | **0.011** | 0.071 | **0.032** | 0.396 | **0.017** | 0.097 | 0.497 | 0.734 | 0.308 | 0.079 | 0.396 | **0.049** | 0.234 |
| Urengoy | **0.043** | 0.634 | **0.047** | **0.028** | **0.033** | 0.053 | **0.023** | **0.047** | **0.034** | 0.367 | **0.044** | **0.045** | 0.475 | 0.684 | 0.324 | **0.034** | **0.034** | 0.157 | **0.048** | **0.034** | 0.480 | 0.077 | **0.031** | 0.180 | 0.087 | 0.089 |
| Tazovskiy | **0.047** | 0.084 | **0.026** | 0.258 | **0.018** | **0.048** | **0.021** | **0.021** | **0.025** | **0.047** | 0.115 | 0.331 | **0.045** | **0.045** | **0.01** | **0.042** | 0.703 | **0.011** | **0.011** | **0.048** | 0.396 | 0.115 | 0.090 | **0.042** | **0.039** | 0.146 |
| Khanymey | | | | | | | | | | | | | | | | | | | | | | | | | | |
| Pangody | 0.061 | 0.253 | **0.013** | **0.013** | **0.007** | 0.067 | **0.005** | **0.015** | **0.006** | 0.794 | **0.005** | 0.731 | **0.004** | **0.024** | **0.004** | 0.143 | 0.243 | **0.01** | 0.113 | **0.042** | **0.047** | 0.559 | 0.516 | 0.330 | 0.144 | 0.746 |
| Urengoy | **0.044** | 0.084 | **0.039** | 0.117 | 0.387 | **0.037** | **0.008** | **0.035** | **0.012** | 0.748 | **0.031** | 0.702 | 0.104 | **0.043** | **0.047** | 0.138 | **0.044** | 0.363 | **0.037** | **0.038** | **0.024** | 0.662 | **0.045** | 0.473 | 0.299 | **0.044** |
| Tazovskiy | **0.017** | 0.076 | **0.049** | **0.037** | **0.022** | **0.026** | 0.612 | **0.017** | **0.019** | **0.028** | 0.089 | 0.612 | **0.004** | **0.046** | **0.002** | 0.316 | 0.075 | **0.002** | **0.033** | **0.043** | **0.005** | 0.394 | 0.073 | **0.045** | **0.043** | 0.374 |
| Pangody | | | | | | | | | | | | | | | | | | | | | | | | | | |
| Urengoy | **0.047** | **0.045** | **0.039** | **0.046** | **0.014** | **0.033** | **0.016** | 0.631 | **0.014** | **0.041** | **0.014** | 0.706 | **0.041** | **0.024** | 0.068 | **0.021** | **0.023** | **0.014** | **0.033** | 0.066 | **0.043** | 0.783 | **0.031** | **0.061** | 0.307 | **0.036** |
| Tazovskiy | **0.016** | **0.036** | **0.039** | 0.408 | **0.004** | **0.021** | **0.028** | 0.169 | **0.044** | 0.364 | **0.002** | 0.537 | 0.396 | **0.048** | **0.001** | 0.076 | 0.347 | **0.043** | **0.045** | 0.280 | 0.280 | 0.643 | 0.440 | **0.044** | 0.67 | 0.077 |
| Urengoy | | | | | | | | | | | | | | | | | | | | | | | | | | |
| Tazovskiy | 0.094 | **0.048** | **0.031** | **0.039** | **0.047** | 0.094 | **0.030** | 0.506 | **0.009** | 0.053 | **0.009** | 0.885 | **0.048** | **0.038** | **0.009** | 0.147 | **0.014** | **0.014** | 0.312 | 0.131 | 0.470 | 0.785 | **0.041** | **0.030** | **0.573** | **0.014** |
| **Hollow** | | | | | | | | | | | | | | | | | | | | | | | | | | |
| Results of all site comparison in the hollow for each component (Kruskal–Wallis H-test) | | | | | | | | | | | | | | | | | | | | | | | | | | |
| Kogalym Khanymey Pangody Urengoy Tazovskiy | H = 12.709 p = 0.013 | H = 9.889 p = 0.04 | H = 9.984 p = 0.041 | H =10.117 p = 0.039 | H =12.937 p = 0.012 | H = 9.871 p = 0.043 | H = 9.865 p = 0.043 | H =13.673 p = 0.008 | H = 9.997 p = 0.041 | H =10.235 p = 0.037 | H = 10.01 p = 0.04 | H =10.112 p = 0.037 | H =10.397 p = 0.034 | H = 9.705 p = 0.046 | H = 15.095 p = 0.005 | H = 9.611 p = 0.048 | H = 9.805 p = 0.044 | H = 11.32 p = 0.023 | H = 15.728 p = 0.003 | H = 9.716 p = 0.046 | H = 10.157 p = 0.038 | H = 9.877 p = 0.043 | H = 10.129 p = 0.038 | H =10.072 p = 0.039 | H =10.119 p = 0.039 | H =10.071 p = 0.039 |
| Pairwise comparison of key sites for each component (Mann–Whitney U test) | | | | | | | | | | | | | | | | | | | | | | | | | | |
| Kogalym | | | | | | | | | | | | | | | | | | | | | | | | | | |
| Khanymey | **0.022** | 0.795 | **0.049** | **0.035** | **0.036** | 0.255 | **0.024** | **0.041** | 0.051 | 0.253 | 0.144 | **0.045** | **0.044** | **0.022** | **0.034** | **0.045** | 0.355 | 0.165 | 0.443 | **0.045** | **0.042** | **0.043** | **0.034** | **0.045** | **0.045** | 0.065 |
| Pangody | **0.016** | 0.343 | **0.048** | **0.016** | **0.038** | 0.643 | **0.034** | **0.016** | **0.035** | **0.007** | 0.143 | **0.029** | **0.014** | **0.041** | 0.155 | 0.096 | 0.305 | **0.024** | **0.034** | 0.355 | 0.065 | **0.024** | **0.014** | **0.034** | **0.032** | **0.024** |
| Urengoy | **0.038** | **0.048** | **0.046** | 0.064 | **0.026** | **0.035** | **0.013** | **0.019** | **0.047** | **0.034** | **0.035** | **0.014** | 0.147 | **0.032** | 0.095 | **0.045** | **0.034** | **0.045** | **0.024** | **0.044** | **0.045** | 0.243 | 0.443 | 0.355 | 0.165 | 0.443 |

| | | | | | | | | | | | | | | | | | | | | | | | | | | |
|---|---|---|---|---|---|---|---|---|---|---|---|---|---|---|---|---|---|---|---|---|---|---|---|---|---|---|
| Tazovskiy | **0.044** | 0.379 | **0.037** | **0.037** | **0.040** | **0.041** | **0.045** | **0.031** | 0.055 | **0.024** | 0.570 | **0.048** | **0.014** | **0.044** | **0.021** | 0.570 | 0.245 | **0.019** | **0.03** | 0.612 | 0.092 | 0.093 | **0.048** | **0.040** | **0.040** | 0.343 |
| Khanymey | | | | | | | | | | | | | | | | | | | | | | | | | | |
| Pangody | **0.028** | **0.046** | **0.035** | **0.046** | **0.037** | 0.149 | **0.021** | **0.036** | **0.016** | **0.025** | **0.021** | **0.024** | **0.048** | **0.038** | 0.364 | **0.048** | 0.564 | **0.021** | **0.039** | 0.149 | 0.573 | **0.033** | 0.248 | 0.248 | **0.033** | **0.032** |
| Urengoy | **0.021** | 0.386 | **0.045** | 0.248 | **0.026** | **0.048** | **0.035** | **0.021** | **0.019** | **0.046** | 0.248 | **0.020** | 0.056 | **0.047** | **0.021** | 0.051 | **0.021** | 0.564 | **0.021** | **0.021** | **0.043** | 0.673 | **0.048** | 0.564 | 0.673 | **0.049** |
| Tazovskiy | **0.037** | **0.026** | **0.042** | 0.085 | **0.023** | **0.048** | **0.041** | **0.026** | **0.048** | **0.047** | **0.045** | **0.046** | 0.057 | 0.085 | **0.008** | 0.571 | 0.695 | **0.038** | **0.014** | **0.048** | **0.048** | 0.750 | 0.062 | 0.345 | 0.450 | 0.705 |
| Pangody | | | | | | | | | | | | | | | | | | | | | | | | | | |
| Urengoy | **0.008** | 0.752 | **0.048** | 0.248 | **0.021** | 0.03 | **0.022** | **0.020** | **0.019** | **0.048** | **0.043** | 0.051 | **0.032** | **0.044** | **0.043** | 0.564 | **0.021** | **0.021** | **0.021** | **0.043** | **0.021** | **0.021** | **0.021** | **0.043** | **0.021** | **0.043** |
| Tazovskiy | **0.020** | 0.449 | **0.047** | 0.055 | **0.008** | 0.014 | **0.020** | 0.185 | **0.045** | **0.043** | 0.245 | **0.046** | **0.048** | **0.048** | **0.008** | 0.105 | 0.624 | 0.145 | **0.014** | 0.571 | **0.023** | **0.038** | **0.048** | 0.571 | **0.049** | **0.038** |
| Urengoy | | | | | | | | | | | | | | | | | | | | | | | | | | |
| Tazovskiy | 0.128 | 0.386 | **0.036** | **0.045** | **0.020** | 0.257 | **0.044** | **0.018** | **0.023** | **0.044** | **0.035** | **0.034** | 0.185 | 0.051 | **0.025** | **0.045** | **0.014** | **0.032** | 0.631 | **0.023** | 0.605 | 0.186 | 0.059 | **0.048** | 0.186 | 0.250 |
| **Permafrost subsidence/frost crack** | | | | | | | | | | | | | | | | | | | | | | | | | | |
| Pairwise comparison of key sites for each component (Mann–Whitney U test) | | | | | | | | | | | | | | | | | | | | | | | | | | |
| Khanymey | | | | | | | | | | | | | | | | | | | | | | | | | | |
| Urengoy | **0.045** | **0.048** | **0.046** | **0.046** | **0.034** | **0.044** | **0.047** | 0.164 | **0.045** | **0.040** | 0.089 | 0.327 | 0.354 | **0.022** | **0.042** | **0.03** | **0.034** | 0.164 | **0.022** | **0.033** | **0.03** | **0.048** | **0.044** | **0.033** | **0.048** | **0.03** |
| Tazovskiy | **0.046** | 0.804 | **0.036** | **0.044** | 0.354 | 0.386 | 0.672 | **0.035** | **0.046** | **0.025** | 0.523 | 0.386 | **0.043** | **0.048** | **0.01** | 0.503 | **0.026** | **0.02** | **0.043** | 0.673 | **0.043** | 0.264 | 0.086 | **0.048** | 0.603 | 0.564 |
| Urengoy | | | | | | | | | | | | | | | | | | | | | | | | | | |
| Tazovskiy | **0.036** | **0.035** | **0.044** | **0.036** | **0.038** | 0.505 | **0.026** | **0.042** | **0.016** | **0.036** | **0.046** | **0.016** | **0.046** | **0.026** | **0.006** | **0.018** | **0.016** | **0.016** | **0.048** | **0.026** | 0.505 | **0.036** | **0.024** | **0.03** | **0.036** | **0.026** |

**Specific comments of Reviewer # 2 (related to line number in the manuscript):**

*15 - "is one of the major consequences" currently unclear.* Should be corrected as "…is expected to enhance under…"
*16 - deepening not "rise"* Agree
*21 - "expected decrease" why would you expect the intensity of DOC and TE mobilisation to decrease? I think you need to provide a rationale for this hypothesis earlier on.* We agree with this remark but it is hard to provide this information in the Abstract. As it is stated in L 347-352, "One can expect that dissolved element decreases its concentration in the peat porewater northward regardless of the micro-landscape due to *i*) decrease of the thickness of peat deposits in total and the active soil (peat) layer in particular (Beilman et al., 2009; Novikov et al., 2009: Stepanova et al., 2015) which decreases the amount of peat interacting with downward penetrating fluids; *ii*) decrease of plant biomass (Frey and Smith, 2007), which diminishes the amount of plant litter that can release the elements (Pokrovsky et al., 2006; Fraysse et al., 2010), and also decrease the plant ability to weather minerals within the soil profile (Moulton et al., 2000); *iii*) shortening the unfrozen period of the year leading to the decrease of the residence time of water in soil pores and *iv*) overall decrease of the intensity of chemical weathering, CO2 consumption and riverine fluxes with mean annual temperature decrease (Dessert et al., 2003)".

*27 - Need to define REEs on first use. What does this stand for (rare earth elements?) and what are you including in this group? actual values ?* The rare-earth elements (REE) include all naturally occurring lanthanides except promethium.

*33-36 "will not exceed 20%" actual values - do you mean they will not change from current values?* - Yes, we do not expect any significant change from current values.
*55- misspelt arctic* - Agree, will be corrected
*65-67 - please check these references with what you are referring too. For example, I do not think Mann et al 2015 examines lakes or Vonk et al. 2015b soil leachates?* - We totally agree and thank the reviewer for this remark. Indeed, Mann et al examined rivers, not lakes. We also corrected the reference of Olefeldt et al and that of Vonk et al. 2015a, not 2015b.

*Also, if you are discussing soil porewaters, you should likely include: "Optical properties and bioavailability of dissolved organic matter along a flow-path continuum from soil pore waters to the Kolyma River mainstream, East Siberia Frey et al. Biogeosciences"* Thank you very much for this valuable reference, which we have missed in our analysis!

*General - should be consistent with use of either trace element or TE throughout.* - Agree and corrected accordingly to "TE" in the main text.

*82 - elements* replaced to TE
*86 - "feeding of" is awkward, maybe use "source to"* Agree with this suggestion
*87 - reference needed for this statement.* - Added Novikov et al. (2009)
*152 - its not clear to me where you used the ANOVA in the results - maybe I missed it?*
Following the recommendation of this reviewer, we performed the non-parametric Kruskal-Wallis test which is suitable for our multi-component data set, as presented above.

*158-162 Here is where I think we need far more info on the approach used in the PCA.* We agree and greatly extended this part of the text as following:
"The PCA analysis allowed to test the influence of various parameters, notably the latitude and the ALT on the soil porewater DOC and element variability. All the variables were normalized as required in standard package of STATISTICA-7 (http://www.statsoft.com) given that the units of measurements of various components are different. The identification of factors was

performed using the method of Raw Data and the extraction method was principal component. The scree test involved plotting the eigenvalues in descending order of their magnitude against their factor numbers and determining where they level off. The PCA values demonstrated significant decrease of the value between F2 and F3 suggesting therefore that at least two factors are interpretable."

*161 - I don't think it really acts upon (as this suggests its constrained in some way) rather it 'explains' a greater variance in.* - Agree and corrected accordingly

*162 - Need more information here on how you used the PCA to test the influence of lat and ALT on DOC and TE.*
We used factor analysis to better understand the data via distinguishing the cluster structure, separation of data sets on similar groups and identifying the groups of elements exhibiting similar distribution pattern. For this, we run the Factor Analysis, Principal Components and Classification Analysis. For determination of the number of variables used for evaluation of element concentration pattern in the data sets and computing the degree of similarity between the elements we used the Cluster Analysis.

*Were you planning to relating the PC loadings or running constrained PCAs?*
This is certainly very good idea. Running the CPCA (i.e., Yoshio Takane «Constrained Principal Component Analysis and Related Techniques») which combines both regression analysis and PCA could provide new view of the multi-componental soil pore water data; unfortunately, we could not realize this approach on our software resources. We do plan to run such analysis using MATCAD and simultaneously analyze, across the same latitudinal profile of the WSL, soil solutions (this study), atmospheric aerosols (Shevchenko et al., 2016 HESS in review), river water (Pokrovsky et al., 2016b), lake water (Manasypov et al., 2014, 2016) and peat elementary composition (Stepanova et al., 2014). However performing such a comprehensive analysis goes beyond the scope of the present work.

*168 - Did you normalize and standardize each of the measurements? This can have a dramatic effect upon PCA loadings and the potential weighting effects of each measurement.* The identification of factors was performed using the method of Raw Data and the extraction method was principal component. All the variables were normalized as necessary in standard package of Statistica-7 given that the units of components are different. The variation method was Varimax Raw because it efficiently minimizes the number of variables having high factor loading.
We have also attempted Principal Components and Classification Analysis. The PCCA yielded the same factor structure but less representative dispersion diagram as shown below in **Fig 2R**.

*169 - I don't think it really acts upon (as this suggests its constrained in some way) rather it 'explains' a greater variance in.* - Agree and corrected accordingly.

*173-176 - Does this mean all of the these were significant to «0.05 and have greater R value of > 0.5?* - Yes, this is true.
*In associated supplemental – can you example what the colored arrows on this plot refer to?* The colored arrows on this plot refer to 6 different group of elements.
*and what W stands for?* W stands for tungsten. It does not exhibit any clear link to other elements. We interpret this behavior as due to important atmospheric loading of W, Cd and B as confirmed by mass balance analyses of atmospheric snow deposition in the WSL (Shevchenko et al., 2016). As a result, these elements are not influenced by intra-soil processes and not affected by mobilization either from peat or from underlying mineral deposits.

[Figure]

Fig. 2R. Element dispersion diagram using Principal Components and Classification Analysis.
Eigenvalues: 3,02341  1,29455  1,07612  ,955973  ,860014  ...

*206 - this is unclear to me. So you did separate tests for each possible site? I think you should run one capable of testing for overall differences first and then examining significant differences.* This is exactly what has been done. We understand that the reviewer is confused. The way it was written in original text was unclear. We revised this sentence as following: "In order to examine the latitudinal trend of element concentration in the porewater, first we run the Wilcoxon-Mann Whitney and Kruskal-Wallis tests for overall differences. After that we assessed, which micro-landscape exhibited the largest difference between sites."

*216 - So why show linear regressions? I would only add these if there is are significant differences between at least the two end member sites. Adding them to all of the graphs just makes its harder to see the values and error bars.* We totally agree and revised Fig 3, 4, 5, S4 and S5. We kept only the correlations that were statistically significant and removed all the lines and equations in the plots where no statistically significant link between the element concentration and latitude was observed. The revised figures (3, 4, 5, S4 and S5) are given in the end of this reply.

*Fig S3 figure text needs an explanation of the circled areas of A.* Two circled areas on Fig S3A correspond to two factors separated by PCA treatment. The first factor explains a greater variance in heavy element hydrolysates such as REEs, Cr, Nb, Zr, Hf, Th and U whereas the second factor was pronounced for soluble and biogenic elements (Mn, Co, Ni, V, Si, Ca, Mg, Sr), pH and latitude but also included Al and Fe, presumably due to organic complexation.

*Also, are both not containing the same explanatory variables and neither showing site loadings?* The PCA loading map is shown in Fig. S3 B. We did not completely understand this question.

*228 - does this mean that R-values varied between these? This way of showing the range is unusual to me.* Yes, the R- values ranged from 0.45 to 0.62 which signifies statistically significant correlation and that is what we aimed to illustrate.

*243 - were these also SUVA 280? As most studies use SUVA at 254 nm.* The UV absorbance of the filtered samples was measured at 280 nm using quartz 10-mm cuvette on Cary-50 spectrophotometer. The specific UV-absorbency at 280 nm ($SUVA_{280}$, L mg$^{-1}$m$^{-1}$) is used as a proxy for aromatic C, molecular weight and source of DOM (Uyguner and Bekbolet, 2005; Weishaar et al., 2003; Ilina et al., 2014 and references therein). The main reason of using $SUVA_{280}$ instead of $SUVA_{245}$ or $SUVA_{254}$ in the present study is for consistency with numerous previous measurements of lakes and rivers in western Siberia (Shirokova et al., 2013; Manasypov et al., 2015, 2017; Pokrovsky et al., 2015) and permafrost-draining rivers in Central Siberia (Prokushkin et al., 2011). More importantly, there is a strong and linear relationship between the absorption at various UV-range wavelength in western Siberian surface waters as shown in **Figure 3R** below. Overall, we believe that the $SUVA_{280}$ can adequately represent the optical properties of DOM in WSL peat porewaters.

[Figure]

**Fig. 3R.** A linear correlation ($R^2 = 0.998$) between UV absorbency at 245 and 280 nm in surface waters of WSL rich in DOC.

*249-256 - I think a range of 2 to 3.5 for SUVA is actually very large. For SUVA 254 for example, we may only expect a natural variation of between 1.5 to 5.5 in pore to coastal waters (in Eastern Siberian freshwaters). A change from 2 to 3.5 demonstrates a significant shift in the composition of the DOM and will have a pronounced effect upon the biogeochemical processing of DOM upon export.* Here, we totally agree with the reviewer that the change of SUVA from 2.4 to 3.4 in hollows shown in Fig 4A demonstrates a significant shift in the composition of the DOM and may have a pronounced effect upon the biogeochemical processing of DOM upon export and we thank the reviewer for pointing out important findings of Frey et al. (2016) which will be cited in the text.

*Leachates of permafrost and active layer peats will also demonstrate clearly that SUVA254 at least id much lower in permafrost material across at least most parts of Siberia and Alaska that have been studied. Would the values you have collected not simply be indicative of collecting waters predominately sources from active layer soils and limited permafrost thaw influence?* We also agree with this proposition. However, the analogy between relatively "fresh" peat soil of Western Siberia (1-2 ky) and old organic matter (8-12 ky) in yedoma of Eastern Siberia is not straightforward.

*Could the higher SUVA further North not instead be suggestive of lower rates of C processing within soil environments?* Yes, statistically significant increase of SUVA$_{280}$ northward in hollows (R$^2$ = 0.599, see Table 3) may indicate the lower rates of DOM processing in soils in the north, linked to either shorter residence time of soil fluids or weaker processes of photo- and bio-degradation in continuous permafrost zone compared to sporadic and discontinuous zone.

*399 - "incoming into" may be instead "prior to export to"* Agree and corrected.

[Figure]

**Figure 3.** Mean values of Specific conductivity (A), pH (B), DOC (C), DIC (D), $SO_4^{2-}$ (E), Si (F), Fe (G) and Ti (H) in peat porewaters of the WSL as a function of latitude for mound and polygons (solid diamonds), hollow (open diamonds), frost crack (grey triangles) and permafrost subsidence/depression (hatched circles). The solid line is a linear fit to all data with the regression equation given on each graph.

[Figure]

**Figure 4.** Mean values of SUVA280 (A), Mg (B), Ca (C), Al (D), Ti (E), V (F), Ni (G) and Sr (H) in peat porewaters of the WSL as a function of latitude for mound and polygons (solid diamonds), hollow (open diamonds), frost crack (grey triangles) and permafrost subsidence/depression (hatched circles). The solid line is a linear fit to all data with the regression equation given on each graph.

[Figure]

**Figure 5.** Mean values of Cl (A), Zn (B), Cd (C), Pb (D), Sb (E) and Rb (F) in peat porewaters of the WSL as a function of latitude for mound and polygons (solid diamonds), hollow (open diamonds), frost crack (grey triangles) and permafrost subsidence/depression (hatched circles). The solid line is a linear fit to all data with the regression equation given on each graph.

[Figure]

**Figure S4.** Mean values of K (A), Na (B), B (C), Li (D), Cr (E), Ba (F), Mo (G), As (H), La (I), Ce (J), U (K) in peat porewaters of the WSL as a function of latitude for mound and polygons (solid diamonds), hollow (open diamonds), frost crack (grey triangles) and permafrost subsidence/depression (hatched circles). The solid line is a linear fit to all data with the regression equation given on each graph.

[Figure]

**Figure S5.** Mean values of Mn (A), Co (B), Zr (C), Hf (D), Yb (E), Th (F), Cs (G) in peat porewaters of the WSL as a function of latitude for mound and polygons (solid diamonds), hollow (open diamonds), frost crack (grey triangles) and permafrost subsidence/depression (hatched circles). The solid line is a linear fit to all data with the regression equation given on each graph.

---

## Author Comment (AC2) · 6 May 2017

The reviewer appreciated substantial amount of data, the reader-friendly presentation by using a clear text structure and good statistical techniques. He/she issued the following minor suggestions.

*-Abstract: I would propose to either replace the final sentence, or add another one, something along the lines of line 510-512 (conclusions) to create an ending that is a bit more general.* We agree and added "The decrease of DOC and metal delivery to small rivers and lakes by peat soil leachate may also decrease the overall export of dissolved components from continuous permafrost zone to the Arctic Ocean"

*- line 101: you here write that the precipitation gradient is from 400 to 460 mm but in Table 1 it ranges between 363 and 594 mm?* We corrected as "The annual precipitation ranges from 600 mm in Kogalym to 360 mm in Tazovsky".

*-is it possible to add one-two lines on the difference in origin for the two micro-landscapes you sketch out in Figure 2?* The initial bog with weakly pronounced micro-relief was subjected to freezing during Subboreal period (~ 4500 y.a). During Subatlantic period (2500 y.a.) and the increase of temperature and precipitation, the thermokarst started. The hollows received sufficient water and they started to thaw, whereas the mounds were rising due to ice wedges underneath (Ponomareva et al., 2012; Panova et al., 2010; Pastukhov et al., 2016).

Ponomareva, O. E., Gravis, A. G., and Berdnikov, N. M.: Contemporary dynamics of frost mounds and flat peatlands in north taiga of West Siberia (on the example of Nadym site), Kriosfera Zemli, XVI, № 4, 21–30, 2012.

Pastukhov, A. V., Marchenko-Vagapova, T. I., Kaverin, D. A., and Goncharova, N. N.: Genesis and evolution of peat plateuas in the sporadic permafrost area in the European North-East (middle basin of the Kosyu river), Earth's Cryosphere, XX(1), 3–13, http://www.izdatgeo.ru/pdf/earth_cryo/2016-1/3_eng.pdf, 2016.

Panova N. K., Antipina T. G., Gilev A.V., Trofimova S. S., Zinoviev E.V., and Erokhin N. G. Holocene dynamics of vegetation and ecological conditions in the Southern Yamal Peninsula according to the results of comprehensive analysis of a relict peat bog deposit, Russ. J. Ecol., 41(1), 20–27, DOI: 10.1134/S1067413610010042, 2010.

*- section 2.2: can you add some references for this method?* The chemical composition of interstitial soil solution is known to depend on the extraction method (e.g., Geibe et al., 2006; Schlotter et al., 2012). Detailed comparison between suction cup and press techniques is described in methodological work of our group (Raudina et al., 2016)

Geibe, C.E., Danielsson, R., van Hees, P.A.W., Lundström, U.S.: Comparison of soil solution chemistry sampled by centrifugation, two types of suction lysimeters and zero-tension lysimeters, Appl. Geochem., 21(12), 2096-2111, doi: 10.1016/j.apgeochem.2006.07.010, 2006.

Raudina, T.V., Loyko, S.V., Krickov, I.V., Lim, A.G.: Comparing the composition of soil waters of West Siberian frozen mires sampled by different methods, Vestnik Tomskogo gosudarstvennogo universiteta. Biologiya – Tomsk State University Journal of Biology, 3(35), 26-42. doi: 10.17223/19988591/35/2, 2016.

Schlotter, D., Schack-Kirchner, H., Hildebrand, E.E., von Wilpert, K.: Equivalence or complementarity of soil-solution extraction methods, J. Plant Nutr. Soil Sci., 175(2), 236-244, doi: 10.1002/jpln.201000399, 2012.

*- line 147-148: how many of the analyses did not show a good agreement?* Just a few 'volatile' trace elements such as Ge, Se, Nb, Sn, Te could not be measured by quadrupole ICP MS. The analysis of these elements require totally different technique.

*- lines 215-216: did you also consider comparing latitudinal gradients for mounds only, or for hollows only (instead of the average values per site independent of topography)?* We did compare the gradient separately for hollows, and for mounds/polygons. Results of linear

regression are listed in Table 3 and described in the end of section 3.3. For example, the most pronounced trend of element concentration increase northward was observed in mounds for Al ($R^2 = 0.91$), Sr ($R^2 = 0.69$), Zr ($R^2 = 0.57$), Ce ($R^2 = 0.76$), Hf ($R^2 = 0.68$) and Th ($R^2 = 0.92$). For these elements, the trend in hollows/cracks was much less pronounced or even absent, with $R^2 < 0.5$ (Table 3). A decreasing trend of element concentration northward was also better pronounced in mounds for Na, Cl, Rb, Cs and Pb.

*- section 4.1: I am wondering: can the difference in DOC mounds vs. hollows also somehow be related to the (seasonal) timing of thaw? (Do the mounds thaw later than the hollows?) And hence the period of unfrozen exchange of constituents in the soil with porewater?* The reviewer is totally right: the mounds do thaw later than the hollows, because the ALT deepens slower (c.a., a factor of 2) at the former. This is linked to both lower amount of heat that is delivered to the mounds with water and to stronger freezing of mounds is winter. However, the water resides longer in mounds than in hollows because the water flow rate through the hollows is a factor of 10 to 20 faster (Novikov et al., 2010).

*Also, in line 379 you briefly mention that the chemical composition of peat between hollows and mounds may be different and could cause the differences in major and TE. Can this different chemical composition of peat not also play a role for the difference in DOC content between mounds and hollows?* The organic carbon in peat of WSL is independent on the micro-landscape and latitude so we do not expect that chemical composition may be important here. In contrast, the peat structure, texture, hydraulic conductivity as discussed in L 288-294 certainly play very important role. Comparison of chemical composition of peat between mounds and hollows across the full latitudinal gradient goes beyond the scope of the present manuscript which is already quite long and will be a subjected of separate publication

*- line 257-259: this is an interesting statement and reference, but could you elaborate a bit more on how this relates to the above two sentences?* The change of SUVA from 2.4 to 3.4 in hollows demonstrates a significant shift in the composition of the DOM and may have a pronounced effect upon the biogeochemical processing of DOM upon export as it has been recently shown in Eastern Siberia (Frey et al., 2016). In the present study, statistically significant increase of $SUVA_{280}$ northward in hollows ($R^2 = 0.599$, see Table 3) may also indicate the lower rates of DOM processing in soils in the north, linked to either shorter residence time of soil fluids or weaker processes of photo- and bio-degradation in continuous permafrost zone compared to sporadic and discontinuous zone.

Frey, K. E., Sobczak, W. V., Mann, P. J., and Holmes, R. M.: Optical properties and bioavailability of dissolved organic matter along a flow-path continuum from soil pore waters to the Kolyma River mainstem, East Siberia, Biogeosciences, 13, 2279-2290, doi:10.5194/bg-13-2279-2016, 2016.

*- line 325-328: if DOC, Fe and Al are dominating colloidal carriers, why do none of the trace elements correlate to DOC?* Good point. This is also observed in river waters draining WSL peatlands. The DOC and Fe are not correlated in rivers (Pokrovsky et al., 2016a) and this is consistent with decoupling of Fe and DOC as two independent colloidal pools (high molecular weight Fe, Al-rich and low molecular weight $C_{org}$-rich), already demonstrated for European rivers (Neubauer et al., 2013; Vasyukova et al., 2010) and other Siberian rivers and WSL thermokarst lakes (Pokrovsky et al., 2006; Pokrovsky et al., 2011, 2016b).

Neubauer, E., Kohler, S.J., von der Kammer, F., Laudon, H., and Hofmann, T.: Effect of pH and stream order on iron and arsenic speciation in boreal catchments, Environ. Sci. Technol., 47, 7120-7128, 2013.

Pokrovsky, O. S., Shirokova, L. S., Kirpotin, S. N., Audry, S., Viers, J., and Dupré, B.: Effect of permafrost thawing on the organic carbon and metal speciation in thermokarst lakes of western Siberia, Biogeosciences, 8, 565-583, 2011.

Vasyukova, E.V., Pokrovsky, O.S., Viers, J., Oliva, P., Dupré, B., Martin, F., and Candadaup, F.: Trace elements in organic- and iron-rich surficial fluids of the boreal zone: Assessing colloidal forms via dialysis and ultrafiltration, Geochim. Cosmochim. Acta, 74, 449-468, 2010.

*- lines 330-336: you present quite a lot of specific information/knowledge here, can you provide a bit better explanation so that more readers can follow?* We agree and revised this paragraph as following: "There are two possible sources of "lithogenic" elements in the peat and peat porewaters: atmospheric dust deposition at the moss and lichen surface and upward migration of soil fluids that carry mineral particles from underlying loam horizons. The loam horizons are rich in silicate clay minerals (e.g., Ovchinnikov et al., 1973; Golovleva et al., 2017) that contain insoluble elements. The geochemical analysis of TE distribution in WSL peat cores across the studied permafrost gradient allowed to distinguish several categories of TE depending on their source such as soluble atmospheric aerosols, atmospheric dust, underlying mineral layers, plant biomass, surface water flooding (Stepanova et al., 2015). The atmospheric deposition of lithogenic elements in the form of soluble aerosols on the moss surfaces followed by incorporation into the peat is expected to be low as shown by thorough snow analyses across the large WSL gradient (Shevchenko et al., 2016). Therefore, atmospheric dust seems to be the main source of insoluble metals in WSL peat as it is also known from other northern bogs (Shotyk et al., 2016). Regardless of the origin of lithophile elements, we hypothesize that the leaching of insoluble trivalent and tetravalent hydrolysates ($TE^{3+}$, $TE^{4+}$) from solid phase to interstitial soil solution may be restricted by the availability of silicate clay minerals within the peat core."

Golovleva, Yu. A., Avetov, N. A., Bruand, A., Kiryushin, A. V., Tolpeshta, I. I., Krasil'nikov, P. V.: Genesis of taiga poorly differentiated soils in West Siberia, Lesovedenie, № 2, 83–93, 2017, http://lesovedenie.ru/index.php/forestry/article/view/983

Shotyk, W., Bicalho, B., Cuss, C. W., Duke, M. J. M., Noernberg, T., Pelletier, R., Steinnes, E., and Zaccone, C.: Dust is the dominant source of "heavy metals" to peat moss (*Sphagnum fuscum*) in the bogs of the Athabasca Bituminous Sands region on northern Alberta, Environ. Internat., 92-93, 494-506, 2016.

*- lines 378-384: the difference in peat chemical composition is an important point, can you elaborate on this a bit more, also with respect to DOC patterns?* We do not believe that the chemical composition of peat may affect the DOC level in porewaters: the peat is highly uniform in $C_{org}$ level across the gradient and among the micro-landscapes. The organic carbon content in peat of WSL is independent on the micro-landscape and latitude (Kremenetski et al., 2003) so we do not expect that chemical composition may be a governing factor of DOC enrichment in porewaters. In contrast, the peat structure, texture, hydraulic conductivity as discussed in L 288-294 certainly play very important role. Using an analogy of ground surface and deep peat for comparison between negative and positive forms of microrelief, we suggest that the dense peat on mounds and polygons has the pores that are significantly smaller with less interconnection, which leads to more restricted flow and greater turtuosity (Rezanezhad et al., 2009, 2010, 2016). This should increase the water residence time in pores of peat in mounds relative to hollows and allow for efficient enrichment of peat porewater by DOC in the former. Comparison of chemical composition of peat between mounds and hollows across the full latitudinal gradient goes beyond the scope of the present manuscript and will be a subjected of separate publication (in progress).

*- line 440-446: this is also an interesting paragraph, that I think you can expand a bit more. E.g., what can be the consequences of the correction for general (upscaling) calculations that are now made in literature?* Today, the majority of Ca, Mg and $HCO_3^-$ ions carried by rivers are used for calculation the $CO_2$ uptake flux due to chemical weathering, i.e., reaction of atmospheric $CO_2$ with alumosilicate and mafic minerals (Dessert et al., 2003; Beaulieu et al., 2012). Not more than 10% of total riverine flux of Ca, Mg and $HCO_3$ is considered to be due to atmospheric input. The present study demonstrates that in case of small rivers draining WSL frozen peatlands, such corrections should be much higher, up to 80% of total flux. The global consequence of this correction is that the continental-weathering $CO_2$ sink in northern peatland regions might be a factor of 2 to 4 smaller than that is currently deduced from the fluxes of large rivers.

Dessert C., Dupré B., Gaillardet J., Francois L., Allegre C.J.: Basalt weathering laws and the impact of basalt weathering on the global carbon cycle. Chem. Geol., 202, 257–273, 2003.

*- line 467-468: I do not understand why the share of spring runoff from the mounds to rivers and lakes will decrease?* Because the degradation of peat mounds and polygons will be accompanied by the spreading of hollows and depressions under on-going climate change (Pastukhov and Kaverin, 2016), the spring runoff from the mounds to rivers and lakes will decrease.

*And, perhaps related to this, have you considered any future changes in precipitation patterns and/or general wetting/drying of the region?* The permafrost boundary shift and the change of microlandscapes are considered to occur regardless of the change of precipitation. As a first approximation, we assume no change in precipitation, evapotranspiration and riverine runoff in the northern part of WSL (60-68°N), given that the drying trend will be pronounced only in the regions located to the south of 60°N (Alexandrov et al., 2016).

Alexandrov, G. A., Brovkin, V. A., and Kleinen, T. : The influence of climate on peatland extent in Western Siberia since the Last Glacial Maximum, Sci. Reports, 6, 24784, doi:10.1038/srep24784, 2016.

We believe that detailed analysis of the future precipitation patterns and wetting/drying regime in the WSL without taking into account the evapotranspiration by mosses and lichen is impossible and this goes beyond the scope of this manuscript.

*- line 474-476: here you present two scenarios that are presented as (i) OR (ii), but isn't it much more likely that both (i) AND (ii) will occur?* This is very good point and we thank the reviewer for pointing this out. Combining both scenario of permafrost thaw (northward permafrost boundary shift and extending the hollows over mounds) suggests that over the first decades, relatively fast permafrost coverage shift will not be accompanied by the change of micro-landscapes and thus the overall decrease of DOC and metal concentration in peat porewaters will be around 20 to 30%. The average rate of peat formation in Siberian flat-mound bogs is 0.24 mm $y^{-1}$ (Inisheva et al., 2013). Taking into account the climate warming and accelerated peat growth, after 500 to 1000 years which are necessary to form the new ca. 20-cm peat layer, the second scenario will take over and thus up to 2-fold cumulative element concentration decrease in soil fluids of continuous permafrost zone may occur.

Inisheva, L. I., Kobak, K. I., Turchinovich, I. E.: Evolution of the paludification process, and carbon accumulation rate in bog ecosystems of Russia, Geography Natural Resources, 34 (3), 246–253, doi:10.1134/S1875372813030086, 2013.

*- line 481: you write "proportion of mounds between 20 and 50%", is that a proportion of the total landscape? Or a proportion of the total elements? Please explain.* This is the proportion of the total terrestrial landscape. The territory includes frozen bogs composed of mounds (hummocks) and hollows/depressions and thermokarst lakes. The typical proportion of mounds in the terrestrial landscape of the WSL (35±15%, Novikov et al., 2009 and authors' unpublished data). Specifically, we calculated the micro-landscape forms at the Khanymey site as following: lakes, 53%; mounds, 23%; hollows, 10% and depressions, 14%. Without considering lakes, the mounds, hollows and depressions occupy 49, 21 and 30% of the territory, respectively. Detailed discussion of possible evolution of the micro-landscape will be a subject of another publication.

*- line 490-492: the fact that this study contradicts a dominating paradigm is something that can come forward a bit more, in my opinion, such as in the conclusions and/or in the abstract.* We agree on this remark: the dominating paradigm of the increase of DOC, DIC, major cation and metal export fluxes upon the on-going climate warming in boreal and subarctic regions should be revised for the case of frozen peatlands.

*- is there a reason why you measured SUVA$_{280}$ and not the more commonly used SUVA$_{254}$?* The UV absorbance of the filtered samples was measured at 280 nm using quartz 10-mm cuvette on Cary-50 spectrophotometer. The specific UV-absorbency at 280 nm (SUVA$_{280}$, L mg$^{-1}$ m$^{-1}$) is used as a proxy for aromatic C, molecular weight and source of DOM (Uyguner and Bekbolet, 2005; Weishaar et al., 2003; Ilina et al., 2014 and references therein). The main reason of using SUVA$_{280}$ instead of SUVA$_{245}$ or SUVA$_{254}$ in the present study is for consistency with numerous previous measurements of lakes and rivers in western Siberia (Shirokova et al., 2013; Manasypov et al., 2015, 2017; Pokrovsky et al., 2015) and permafrost-draining rivers in Central Siberia (Prokushkin et al., 2011). More importantly, there is a strong and linear relationship between the absorption at various UV-range wavelength in western Siberian surface waters as shown in **Figure 3R** of our reply to **Reviewer No 2** (http://www.biogeosciences-discuss.net/bg-2017-24/bg-2017-24-AC1-supplement.pdf). Overall, we believe that the SUVA$_{280}$ can adequately represent the optical properties of DOM in WSL peat porewaters.

*Tables and figures: - Table 1: write "latitude" instead of "GPS", and perhaps add the abbreviations for the regions (Tz, Ur, etc.) behind the site names.* We agree and corrected accordingly.

*- Figure 1: I think the panel with the actual map can be improved for increased readability, for example: enlarge picture, add either a vegetation map or biome map, or permafrost zonation map (instead of red lines) on the background (instead of the currently-used rather vague colours). Additionally, is it possible to add site maps with more detailed, high-res sampling locations of the different samples?* We greatly revised the maps in Figure 1 following this important remark as shown in Fig 1R of this reply. Now this figure includes the biome boundaries and permafrost zonation map. However, adding detailed sampling locations at the key sites would greatly overload this paper. Besides, the sampling was performed in a relatively small area in each site, which is much better shown via actual aerial images in Fig. 1 than via rather complicated topographical maps. Finally, high–resolution (1:25,000-1:10,000) maps necessary for showing out study sites of these territories are simply not yet available.

*- Figure 2: What is the vertical white line (with a dashed line in it) that crosses panel B through the left polygon?* This vertical line indicated a discontinuity of hydrological flow-path.

- Figure 3, 4, and 5: write "linear" instead of "liner". Also, it may be good to indicate the boundaries between the sporadic-discontinuous and discontinuous-continuous permafrost zones with vertical thin dashed lines? Following this important advice, we revised Figure 3, 4 and 5 as shown in our reply to **Reviewer No 2** (http://www.biogeosciences-discuss.net/bg-2017-24/bg-2017-24-AC1-supplement.pdf)

*Text edits/spelling:*

 *- Title: write "elements" instead of "element"?* Agree and corrected

*- line 55: "arctic" line 156: "landscapes"* Corrected

*- line 211: "pore waters"* Corrected

*- I personally think ALT "rise" is not an ideal way of putting it, I would prefer to use ALT deepening or ALT thickening.* We agree that it is better to say "ALT deepening" or "ALT thickening" and corrected this term throughout the manuscript

*Line 305-307: add "respectively" after this sentence* Corrected

*- line 440: I suggest to write "our obtained results"* Corrected

- line 450: "in accordance" Corrected

- line 464 and 466: "on the one hand" and "on the other hand" (not "from") Corrected

*- Olefeldt should be spelled throughout the manuscript with "dt"* Corrected throughout the manuscript

*In general, the language is quite good but I think the manuscript can benefit from a quick native-speaker check because particularly the use of articles ("the" and "a) is often left out where it is required, and sometimes vice versa.* We carefully proofread manuscript for English grammar and spelling and took into account all grammar remarks of both reviewers. Please note that English corrections are included in Biogeosciences service of open access articles should this paper be considered for publication in BG.

[Figure]

**Figure 1R.** Map of the study site with permafrost boundaries (Brown et al., 2001; http://portal.inter-map.com (NSIDC)), with 5 main test sites: Kogalym (Kg), Khanymey (Kh), Pangody (Pg), Urengoy (Ur) and Tazovsky (Tz). The mean annual temperatures are given in parenthesis. The inserts represent aerial (drone-made) photos of main sites with the position of mound/polygon (M/P), hollow (H), frost crack (FC) and permafrost subsidence (Ps). On the Kogalym site, a hollow (H) – ridge (R) – lake complex is dominating landscape type.

The numbers on the legend represent the following: 1, tundra; 2, forest-tundra; 3, northern taiga; 4, middle taiga; 5, borders between natural biomes; 6, borders between permafrost zones; 7, continuous permafrost; 8, discontinuous permafrost; 9, sporadic permafrost; 10, isolated permafrost; 11, key study sites with mean annual temperature is in the parentheses.